

# Modeling macropore seepage fluxes from soil water content time series by inversion of a dual permeability model

Nicolas Dalla Valle[1], Karin Potthast[2], Stefanie Meyer[3], Beate Michalzik[2], Anke Hildebrandt[4], and Thomas Wutzler[1]

[1]Max Planck Institute for Biogeochemistry, Department Biogeochemical Integration, Hans Knöll-Straße 10, 07745 Jena, Germany
[2]Institute of Geography, Soil Science, Friedrich Schiller-University Jena, Löbdergraben 32, 07743 Jena, Germany
[3]Medical Center LMU Munich, Ziemssenstraße 1, 80336 München, Germany
[4]Institute of Geosciences, Ecological Modelling, Friedrich Schiller-University Jena, Burgweg 11, 07749 Jena, Germany

*Correspondence to:* Nicolas Dalla Valle (ndalla@bgc-jena.mpg.de)

**Abstract.** Dual permeability models are widely used to simulate water fluxes and solute transport in structured soils. However, so far obtaining necessary data for model calibration is a problem due to the large set of unconstrained parameters. Therefore, this study presents a simplified 1D dual permeability model whose structure is similar to the MACRO model together with a calibration scheme that allows constraining the parameters using time series of soil water content. The inversion scheme

consists of four consecutive steps: First, the parameters of three different water retention functions were assessed using vertical soil water content profiles assuming hydraulic equilibrium. Second, the soil sorptivity and diffusivity functions were estimated from Boltzmann-transformed soil water content data of a drying period. Third, the parameters governing macropore flow were determined using the most dynamic part of the soil water content time series during the first 12 h after a precipitation event.

The model was calibrated using data of artificial, homogeneous and shallow soils from mesocosms. The resulting retention

functions predicted similar values as pedotransfer functions apart from for very dry conditions. The predicted soil water content time series were in good agreement with measurements at 5 and 12 cm soil depth. Predicted macropore seepage fluxes exhibited high uncertainty and differed between water retention functions, but average predictions were close to measurements for two of the three water retention functions.

The study demonstrates the feasibility of calibrating a 1D dual permeability model with soil water content time series.

# 1   Introduction

The ability of soils to filter dissolved or particulate matter from percolating water is an important ecosystem service (Costanza et al., 1997). This filtering relies on water moving comparatively slowly and uniformly through the soil matrix where organic and inorganic solutes and particles can be adsorbed to minerals or organic matter and become depolymerized and finally mineralized by microorganisms.

However, water is rarely moving uniformly through soils, but likely follows different preferential flow paths depending on soil structure and texture (e.g., Jarvis et al., 2012; Clothier et al., 2008). One of these preferential flow paths is the soil's macro-



pore network, which can allow water to bypass parts of the soil matrix and thus hamper the soil's filtering ability. Therefore, macropore flow through soils may be a pathway for contaminants to reach the groundwater (Jarvis, 2007). On the other hand, it may also provide carbon, nutrients and energy to deeper soil layers or aquifers influencing subsurface ecology (Akob and Küsel, 2011). Furthermore, it may influence the vertical distribution of organic matter in the soil profile by redistribution of dis-

solved and particulate organic matter (Jardine et al., 2006) or contribute to the ecosystem carbon balance by removing organic matter through leaching (Kindler et al., 2011).

However, the heterogeneity of soils and water inputs hampers the prediction and quantification of preferential flow through macropores. Therefore, suitable soil hydrologic models are required to identify conditions, that may lead to macropore flow, but also to enable quantitative predictions of seepages fluxes, solute and particle redistribution and transport through the soil.

Classic soil hydrologic models are based on Richards equation and assume a hydraulic equilibrium within a representative elementary volume of soil (Köhne et al., 2009; Gerke, 2006) - a condition frequently violated in field soils and particularly invalid for macropore flow (Beven and Germann, 2013; Jarvis, 2007).

Therefore, a large amount of research was dedicated to improve the description of water movement in unsaturated soils in soil hydrological models. These were reviewed by Köhne et al. (2009), Gerke (2006), Šimůnek et al. (2003) and others.

A particular class of soil hydrological models that deal with hydraulic non-equilibrium are the two domain models, which usually feature two spatially superimposed domains to describe flow in the soil matrix and in preferential flow paths. (Köhne et al., 2009; Gerke, 2006). Two domain models can be classified as dual porosity models (mobile-immobile), if the water in the matrix domain does not move, or as dual permeability models (DPM), if both model domains allow for water flow (Köhne et al., 2009; Šimůnek and van Genuchten, 2008; Šimůnek et al., 2003). This setup enables DPMs to provide high flexibility in

the description of water flow, since they can use different parameterizations or even entirely different models for both domains (Šimůnek et al., 2003).

Accordingly, DPMs are usually able to simulate preferential flow phenomena and performed well in model comparison studies e.g. on soil columns with macropores (Arora et al., 2011; Köhne and Mohanty, 2005), multi-step outflow (Laloy et al., 2010) and pesticide transport (Köhne et al., 2006), as well as in field studies on tracer transport (e.g., Gerke and Köhne, 2004).

Therefore, it is not surprising that dual permeability models like MACRO (Jarvis and Larsbo, 2012) DUAL (Gerke and van Genuchten, 1993) as implemented in the HYDRUS-1D and 2D/3D softwares (Šimůnek et al., 2016) or the Root Zone Water Quality Model (RZWQM) (Hanson et al., 1998) are increasingly used in porous media hydrology and in agricultural risk assessment.

This wide applicability of DPMs comes with the prize of requiring a comparatively large amount of parameters for the two

flow domains and an exchange term. The estimation of these parameters is usually costly and some of them may be impossible to measure (Köhne et al., 2009; Šimůnek et al., 2003). As a result, parameters of dual permeability models are often estimated by inverse modeling. This requires measurements of state variables such as soil water content or fluxes such as precipitation and seepage, which can be easier to obtain.

Inverse parameterization of soil hydrological models in general and DPMs specifically often fails to constrain the model

parameters (e.g. Carrera and Neuman, 1986). To minimize these problems model inversions often used a combination of




different data types like e.g. pressure head and water content (Jacques et al., 2002), surface temperature and water content (Steenpass et al., 2010) or seepage fluxes and matric potentials at certain depths (Schelle et al., 2013). Other studies fixed some of the required parameters by using pre-described water retention functions (e.g., Werisch et al., 2014) or estimated some of the parameters using pedotransfer functions based on measurements of soil texture classes (e.g., Scharnagl et al., 2011).

While these approaches usually allow for the retrieval of meaningful parameters, the required data are typically not available, hence hampering the application of DPMs. Accordingly, several studies aimed to parameterize soil hydrological models based on smaller data sets. Promising in this regard is soil water content, as it can be measured comparatively easily, accurately and automatically at the local scale using probes, while at larger scales estimates of soil water content can be determined from satellite remote sensing (Petropoulos et al., 2015).

As a result, several studies used field soil water content data from different depths for inverse parameter estimations of single domain models (Le Bourgeois et al., 2016; Over et al., 2015; Schelle et al., 2013; Scharnagl et al., 2011; Wollschläger et al., 2009; Ritter et al., 2003). Remote sensing data were also widely used to retrieve soil hydraulic parameters as has been reviewed by Mohanty (2013). However, using soil water content data in inverse parameter estimation is not straightforward. Scharnagl et al. (2011) found that field observations of soil water content alone were not sufficient to estimate the parameters of the

widely used van Genuchten water retention function (van Genuchten, 1980). They suggested to use higher frequency water content time series gathered shortly after precipitation events and meaningful prior values for the water retention parameters to improve parameter identifiability (Scharnagl et al., 2011).

Hence, there is a need of developing models and inversion schemes that allow for parameter identification using data from more widely available data sets, like soil water content networks (e.g. Bogena et al., 2010).

Accordingly, the aim of this study is:

1. to develop a DPM of low complexity that still captures flow patterns, but whose parameters can be constrained by soil water content time series data

2. to develop a model inversion scheme to constrain model parameters

The model development resulted in a dual permeability model similar to the MACRO model (Jarvis and Larsbo, 2012). We

explore the usage of three alternative simple, *i.e.* few parameter, water retention functions. Further, this study presents a three-step inversion scheme, where the different steps make use of diverse features of the soil water content time series data to constrain the parameters. An application of the model inversion is demonstrated.

## 2   Model description

The model presented in this paper aims providing means for simulating macropore flow in structured soils. It is deliberately

kept simple to reduce the number of parameters and allow their estimation by inverse modeling using limited data. The model is not meant to be an accurate physical model of water flow, but rather a simplified conceptual description and is intended to be used in field applications in the future. The model is set up as a DPM with a macropore and a soil matrix domain (Fig. 1).



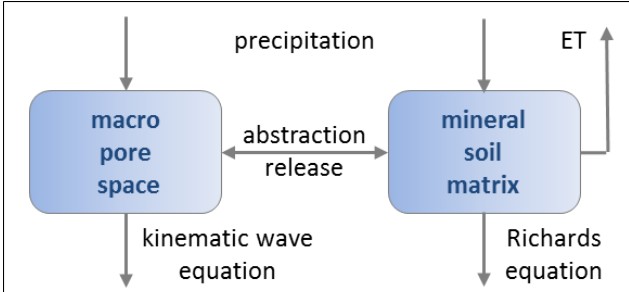

**Figure 1.** Rapid flow model.

Precipitation enters the macropore and matrix domains. Water in the macropore domain is transported by advection and may be abstracted into the mineral soil matrix, where it is transported according to Richards equation. If the matric water content is above field capacity, water can leak out into the macropore domain.

Conceptually, the macropore domain consists of large pores within which water movement is predominantly driven by gravity. These include earthworm burrows and pipes left by decaying roots as well as cracks and fissures. The soil matrix domain on the other hand consists of smaller pores where capillary forces have a significant effect on water movement. Water enters the soil according to the current precipitation rate without ponding. Water in the macropore domain may be abstracted into the matrix domain while moving downward through the soil.

The presented model structure is similar to the MACRO model (Jarvis and Larsbo, 2012) in using the Richards equation for the matrix domain and a kinematic wave equation for the macropore domain. The largest differences are the use of a simplified domain exchange term, the application of different simple retention functions and an indirect simulation of water repellancy in the infiltration and domain exchange terms.

Consult Tables 1 and 2 for a description of symbols and model parameters, respectively.

## 2.1 Macropore domain

Water movement in the macropore domain is described by a kinematic wave approach (Beven and Germann, 1981). This approach assumes water to flow in a film along the macropore walls. The volumetric flux, $q_{ma}$, of such a downward moving water film can be expressed as (Hincapié and Germann, 2009):

$$q_{ma} = F^3 L \frac{g}{3\eta} \tag{1}$$

where $F$ is the thickness of the water film, *i.e.* the distance between the macropore wall and the air-water interface, $L$ is a parameter inversely related to soil macroporosity and defined as $l/A_c$ with $l$ being the contact length between water film and macropore wall and $A_c$ being the macroscopic cross sectional area under consideration, $g$ is the gravitational acceleration and $\eta$ is the kinematic viscosity of water. Assuming a cuboid shape of the water film the macropore water content $W$ can be calculated





as $W = FL$ (Hincapié and Germann, 2009). This can be used to express $q_{ma}$ in terms of $W$ (Hincapié and Germann, 2009):

$$q_{ma}(W) = \frac{g}{3\eta}\frac{1}{L^2}W^3 \tag{2}$$

Differentiating (2) gives the average velocity $v_W$ of the downward moving water film (Hincapié and Germann, 2009):

$$v_W = \frac{dq_{ma}}{dW} = \frac{g}{\eta}\frac{1}{L^2}W^2 \tag{3}$$

Equation (3) is used in an advection equation to determine flow in the macropore domain:

$$\frac{\partial W}{\partial t} = -v_W\frac{\partial W}{\partial z} - A$$
$$\frac{\partial W}{\partial t} = -\frac{g}{\eta}\frac{1}{L^2}W^2\frac{\partial W}{\partial z} - A \tag{4}$$

where A is an abstraction term describing water flow into the soil matrix and t and z are time and the vertical dimension, respectively.

## 2.2  Matrix domain

The matric flow model uses a diffusivity based formulation of Darcy's law (Brutsaert, 2005):

$$q_{mi} = -D(\theta)\frac{\partial \theta}{\partial z} - K \tag{5}$$

where $q_{mi}$ is the matric volumetric flux, $D(\theta)$ is the matric water diffusivity, $\theta$ is the volumetric matric water content and $K$ is the hydraulic conductivity of the soil matrix. This formulation was chosen to avoid using matric potential measurements and to facilitate a model inversion based on soil water content data. The matric water diffusivity $D(\theta)$ can also be expressed as a

function of $K$ and the slope of a water retention function ($\frac{d\psi}{d\theta}$) (Klute, 1952):

$$D(\theta) = K\frac{d\psi}{d\theta} \tag{6}$$

where $\psi$ is the soil matric potential. Equation (6) can be used to replace $K$ in equation (5) resulting in a formulation of Darcy's law that only dependents on the volumetric water content:

$$q_{mi} = -D(\theta)\left(\frac{\partial \theta}{\partial z} + \left[\frac{d\psi}{d\theta}\right]^{-1}\right) \tag{7}$$

Combining this with the continuity equation and the water abstraction from the macropore domain gives a version of the Richards equation that is used in the matric flow model:

$$\frac{\partial \theta}{\partial t} = -\frac{\partial q_{mi}}{\partial z} + A - ET$$
$$\frac{\partial \theta}{\partial t} = \frac{\partial}{\partial z}\left[D(\theta)\left(\frac{\partial \theta}{\partial z} + \left[\frac{d\psi}{d\theta}\right]^{-1}\right)\right] + A - ET \tag{8}$$





where A is an abstraction term describing water exchange between soil matrix and macropore domains and ET is evapotranspiration. The matric water diffusivity varies over several orders of magnitude depending on the volumetric water content. It is calculated as a function of the soil sorptivity (Espejo et al., 2014):

$$D(\theta) = -\frac{S(\theta)}{2c(\theta_s - \theta)} \tag{9}$$

where $S$ is the soil sorptivity (Philip, 1957b), $c$ is a soil-specific parameter and $\theta_s$ is the volumetric water content at matric saturation.

## 2.3 Domain exchange term

A film flow along the macropore walls causes an infinitesimally thin, saturated boundary layer between both domains where $\theta = \theta_s$. The abstraction flux of water from the macropore domain into the matrix domain increases with the ability of the matrix
to transport water away from this saturated layer. In the model this ability is described by the gradient in soil water diffusivity between the saturated boundary layer and the soil matrix. The abstraction flux further increases with the matric saturation deficit $(\theta_s - \theta)$ and the effective saturation of the macropore domain, $S_{ma}$, to account for an incompletely wetted boundary layer (Larsbo et al., 2005). With this reasoning the MACRO model uses the following term to calculate the abstraction flux (Jarvis and Larsbo, 2012):

$$A = S_{ma} \left( \frac{D(\theta) + D(\theta_s)}{2} \right) \frac{G \gamma_w}{d^2} (\theta_s - \theta) \tag{10}$$

where $G$ is a geometry factor, $\gamma_w$ is a scaling factor and $d$ is an effective diffusion path length. Assuming a residual water content of 0 for the macropore domain, the effective macropore saturation can be written as $S_{ma} = \frac{W}{W_s}$ with $W$ and $W_s$ being the macropore water content at current state and at saturation, respectively. Lumping $W_s$, $G$, $\gamma_w$ and $d^2$ into a single parameter $r$ gives the simplified abstraction term used in the model:

$$A = W \left( \frac{D(\theta) + D(\theta_s)}{2} \right) r (\theta_s - \theta) \tag{11}$$

## 2.4 Boundary and initial conditions

The maximum amount of water that can infiltrate into the matrix domain at the soil surface in a given time step depends on the hydraulic conductivity $K_{top}$ of the soil surface. Similar to equation (11) it is assumed that precipitation results in an infinitesimally thin, saturated layer at the soil surface. This allows the calculation of $K_{top}$ as the average of the saturated
hydraulic conductivity and the hydraulic conductivity of the uppermost soil layer using equation (6):

$$K_{top} = \frac{1}{2} \left( D(\theta_s) \left[ \frac{d\psi}{d\theta}(\theta_s) \right]^{-1} + D(\theta_1) \left[ \frac{d\psi}{d\theta}(\theta_1) \right]^{-1} \right) \tag{12}$$

As explained below this formulation was chosen to indirectly simulate effects of water repellency on macropore flow generation. Note that $K_{top}$ is also the maximum possible infiltration rate into the matrix domain for a given water content of the uppermost soil layer. So, up to a precipitation rate of $K_{top}$ water will only enter the matrix domain. For precipitation rates



higher than $K_{top}$, any water above a rate of $K_{top}$ will enter the macropore domain. The upper boundary condition for both domains can therefore be described by a known flux as follows:

$$q_{top_{ma}} = \max\left(0, P - K_{top}\right)$$

$$q_{top_{mi}} = \min\left(P, K_{top}\right)$$

(13)

where $q_{top_{ma}}$ and $q_{top_{mi}}$ are the volumetric fluxes across the upper model boundary into macropore and matrix domain, respectively, and $P$ is the precipitation rate.

For common water retention functions this formulation will result in an exponential decrease of $K_{top}$ with decreasing water content. This is supposed to indirectly simulate the water repellence observed during initial stages of infiltration in many soils (e.g. Ritsema and Dekker (2000), Jarvis et al. (2008)) and allows for the frequently observed initiation of macropore flow at comparatively low precipitation rates in dry soils (Clothier et al., 2008).

In its current form the model allows for free drainage from the macropore domain at the lower model boundary, while no flow is allowed through the lower boundary of the matrix domain. The latter was chosen to reflect our experimental setup (see section 3).

Initial water contents in the macropore domain were set to zero. Initial water contents in the matrix domain were set according to a vertical equilibrium profile based on a water retention function and measurements of volumetric water content. The concept of vertical equilibrium profiles is explained in section 4.1.

## 3 Data for a model test: Mesocosm sprinkling experiments

Data from mesocosm sprinkling experiments provided a testbed under laboratory conditions to explore model calibration.

The circular mesocosms had a diameter of 50 cm, a height of 1 m, a base outlet and an automated sprinkling procedure. They were filled with filling material up to a height of 20 cm. This filling material consisted of 70 % by mass top soil (0 - 16 cm) taken from a temperate pasture site near the Hainich National Park, Thuringia, Germany (ca. 51.1 N, 10.4 E) mixed with 30 % by mass of washed gravel (3 mm in diameter). The fresh top soil was sieved with a mesh size of 6 mm prior to mixing.

Furthermore an aliquot of the filling material was dried at 50 °C, sieved (2 mm mesh size) and used to measure soil texture with a Laser Diffraction Particle Size Analyzer (Beckman Coulter LS 13320, Fraunhofer optical model). In advance, organic substances and carbonate were eliminated from the aliquot by hydrogen peroxide (30 %) and hydrochloric acid (10 %), respectively. According to this, the fine soil consisted mainly of silt particles ($81.7\% \pm 2.9\%$) with minor contributions of clay ($11.2\% \pm 0.8\%$) and sand ($7.1\% \pm 3.3\%$).

The mecosocms were planted with the pasture grass *Dactylis glomerata*. The grass was sown (4 seed dm$^{-2}$) five months before the start of the experiment. The mesocosms were placed in a climate chamber, that ran a constant 8 h dark and 16 h light cycle at 15 °C.

In three mesocosms a total of six soil water content probes (EC-5, Decagon Devices, Pullman, Washington, USA) were installed at two different depths (5 cm and 12 cm) resulting in one probe per depth. The volumetric water content was measured every 10 seconds.





We conducted 15 sprinkling experiments using a rainfall simulator to simulate strong precipitation events. The rainfall simulator used a fixed precipitation rate, but during the experiments precipitation amount and initial water contents were varied. We also measured cumulative seepage fluxes from the mesocosms three days after each precipitation event. The 15 events in three mesocosms resulted in a total of 45 water content time series per depth. However, seepage fluxes were only observed during one event with particularly high precipitation leading to only three water content time series per depth during events with seepage flow.

The high temporal resolution of the soil water content time series data captured rapid dynamics right after the precipitation events, but did not give much information for the remainder of the time series. Therefore, the data were aggregated to make them more manageable and emphasize the most dynamic parts of the time series. A constant depth dependent evapotranspiration rate was estimated form the soil water content time series as well. The exact procedures are described in appendix C.

## 4 Parameter estimation

The model parameters were derived by inverse modeling in *R* version 3.2.5 (R Core Team, 2016) using the *FME* package (Soetaert and Petzoldt, 2010). A first model inversion was done separately for the matric flow parameters, the matric water diffusivity function and the macro flow parameters based on different parts of the data (Tab. 3). The inversion to estimate matric flow parameters and the matric water diffusivity function used data from all available soil water content time series, while the inversion of the macro flow parameters was based only on the time series from the event with observed seepage fluxes. These initial inversion steps provided parameters near the global optimum while minimizing the problem of equifinality and the problem of being trapped in local optima. Accordingly, the resulting parameters were used as starting values in a final inversion that estimated all parameters simultaneously. This final optimization also provided uncertainty distributions of the estimated parameters.

### 4.1 Matric flow parameters

The estimation of matric flow parameters is based on the assumption that vertical water fluxes are negligible before and $12\,\mathrm{h}$ after the sprinkling event, *i.e.* the assumption of hydrostatic equilibrium in the soil profile during those times ($\frac{\partial \theta}{\partial z} = 0$, equation (8)). While a time frame of $12\,\mathrm{h}$ seems short for a natural soil profile, the water content time series only changed very slowly after this period and the mesocosm soils were quite shallow with a depth of $20\,\mathrm{cm}$. We also assume that ET does not affect soil water distribution as it does not considerably affect soil water storage during such short time scales.

Ignoring $A$ and $ET$ and since $D(\theta)$ cannot become $0$ according to equation (8) the vertical profile is in equilibrium, if:

$$\frac{\partial \theta}{\partial z} = -\left[\frac{d\psi}{d\theta}\right]^{-1} \tag{14}$$



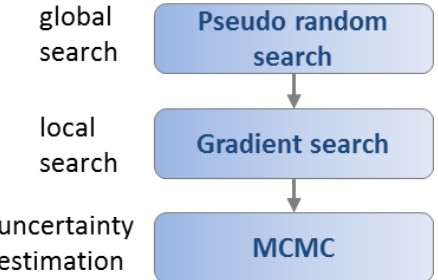

**Figure 2.** Optimization scheme used to estimate matric retention and macropore flow parameters.

After applying global and local search algorithms a Markov Chain Monte Carlo (MCMC) run is done to estimate parameter uncertainty.

Equation (14) can be rearragend and integrated over an arbitrary part of the soil profile from a depth $z$ to a depth $z_{ref}$ with $0 \leq z \leq z_{ref}$ (positive downwards) to yield:

$$\frac{d\psi}{d\theta} d\theta = -dz$$

$$\int_{\theta(z)}^{\theta(z_{ref})} \frac{d\psi}{d\theta} d\theta = \int_{z}^{z_{ref}} -dz \tag{15}$$

$$\psi(\theta(z_{ref})) - \psi(\theta(z)) = z - z_{ref}$$

After inserting a water retention function ($\psi(\theta)$) into equation (15) it can be rearranged to derive a vertical equilibrium profile

function $\theta(z)$. We explored alternative water retention functions: Two exponential approximations and the Campbell function (Campbell, 1974). The functions and their respective vertical equilibrium functions and first derivatives are given in Table 4 and their parameters are explained in Table 2.

The simple exponential approximation (R1) was chosen since it only requires two parameters. Inspired by Walker et al. (2001) we added the modified exponential approximation (R2) despite its more complicated mathematical treatment to also

compare a two parameter retention function with a more realistic shape at low water contents. The Campbell function (R3) was included despite requiring three parameters since it is the special case of the widely used Brooks-Corey function (Brooks and Corey, 1964) where the residual water content, $\theta_r$, is zero, which is also assumed in this study. Also, the additional parameter, $\theta_s$, of the Campbell function is of little consequence (see below).

The vertical equilibrium profile equations in Table 4 show that the vertical distribution of soil water content in a profile at

hydrostatic equilibrium depends on the soil hydraulic properties (*i.e.* the parameters of the retention function) and a reference water content $\theta(z_{ref})$ at a reference depth $z_{ref}$ as an additional parameter. Accordingly, parameters have the same values across all vertical water content profiles except for the reference water contents.

The highest measured water contents occured after the event that exhibited seepage fluxes. We assume that matric water content at the bottom of the mesocosms was close to matric saturation, $\theta_s$, during this event and that the volumetric water

content at matric saturation was equal to matric porosity. We used the volumetric water content at the lower model boundary




as reference since this was already required as a parameter of the Campbell function. Hence, the total number of parameters was the same with the Campbell function.

We then estimated the water retention parameters and the reference water content at the profile bottom as additional parameter for all three water retention functions by minimizing the following objective function:

$$\sum_{i=1}^{n}\left[\left(\frac{\theta_{i_{obs}} - \theta_{i_{pred}}}{\omega_{i_1}}\right)^2 + \left(\frac{\Delta S_{i_{obs}} - \Delta S_{i_{pred}}}{\omega_{i_2}}\right)^2\right]$$

$$\Delta S_{obs} = P - ET - Q_{bot}$$ (16)

$$\Delta S_{pred} = \int\limits_{z=0}^{z=z_{ref}} \theta(z, t=12h) - \int\limits_{z=0}^{z=z_{ref}} \theta(z, t=0)$$

where $n$ is the number of soil water content time series per depth, $\theta_{obs}$ is the observed volumetric water content at the start of the sprinkling experiment and $12$ h after its end at the two measurement depths, $\theta_{pred}$ is the predicted water content at the same times and depths, $\Delta S_{obs}$ is the observed change of water storage in the mesocosms as calculated based on flux measurements, $\Delta S_{pred}$ is the predicted change of water storage calculated as the difference between the integrated soil water content as

estimated using the vertical water content profile functions, $\theta(z)$; $\omega_1$ and $\omega_2$ are weighting factors related to observation uncertainty, $P$ is cumulative precipitation, and $Q_{bot}$ is the cumulative seepage flux from the mesocosms. The weighting factor $\omega_1$ was set to 10 % of the observed water contents and the weighting factor $\omega_2$ was set to 30 % of the observed changes of water storage.

The matrix parameter optimization process is outlined in Figure 2. We first applied 7500 iterations of the global pseudo-
random search optimization algorithm (Price, 1977) as implemented in the *FME* package (Soetaert and Petzoldt, 2010) followed by as many iterations of the (gradient based) quasi-Newton method "BFGS" as implemented in the *optim* function in base R's *stats* package as were necessary to reach convergence.

## 4.2 Estimation of matric water diffusivity function

The matric water diffusivity function $D(\theta)$ was calculated based on a method published by Espejo et al. (2014) using Boltzmann
transformed soil water content data from drying periods starting $12$ h after the start of the precipitation events.

In order to receive a diffusivity function that is valid on the whole water content range, it is important to use data covering as large a range of soil water content values as possible. Since the soil water retention functions and therefore the approximate range of possible water content values for the mesocosm soils were already known from the matrix parameter optimization, it could easily be seen that wet conditions were hardly covered by the data after removal of data points from the period of
$12$ h after the precipitation event. However, soil water content measurements close to saturation are particularly important to estimate the shape of the diffusivity function. Therefore, we extrapolated all soil water content values to the bottom of the soil profile using the three retention functions introduced in section (4.1) and their respective optimized parameters. The extrapolated data gave a better coverage of soil water content close to matric saturation.





The Boltzmann transformation is used to express data that are a function of space - in this case depth $z$ - and time $t$ as a function of a single independent variable defined as $\lambda = zt^{-\frac{1}{2}}$ (Boltzmann, 1894). It is applicable, if the depth $z$ has a similar influence on flow as time $t^{-\frac{1}{2}}$, which requires the problem to meet the following boundary conditions (Brutsaert, 2005):

$$\begin{aligned} \theta &= \theta_i & \lambda &\to \infty & (z \to \infty) \\ \theta &= \theta_0 & \lambda &= 0 & (t \to \infty) \end{aligned} \tag{17}$$

where $\theta_i$ is the inital water content and $\theta_0$ is the end water content. In other words: There must be a point at a large depth where soil water content is almost time constant and after a long time $\theta$ must reach the residual water content for small values of $z$. We assume, that the water content at matric saturation, $\theta_s$, as estimated in matrix parameter optimization (section 4.1) is a reasonable approximation of $\theta_i$ and that the matric water content will reach a residual water content of $0$ after a long time.

These conditions are violated during the early parts of the soil water time series at the beginning of an infiltration event,
which see a rapid rise in water content and where flow still has a significant impact on water content change. Therefore, and according to the assumption stated earlier in section (4.1) that flow did not contribute to soil water content change in the mesocosms at times later than 12 h after the start of the precipitation event, only soil water content measurements obtained after this period and before the next precipitation event were used to estimate the parameters of the diffusivity function.

Under the conditions described above it can be assumed that 12 h after the precipitation event gravity forces are relatively
unimportant in driving water movement compared to capillary forces and equation (8) can be simplified to:

$$\frac{\partial \theta}{\partial t} = \frac{\partial}{\partial z}\left(D(\theta)\frac{\partial \theta}{\partial z}\right) \tag{18}$$

which is equal to the diffusion equation for horizontal matric flow (Brutsaert, 2005). Equation (18) can be changed using the Boltzmann transformation and re-arranged to give a description of the soil water diffusivity (Philip, 1957a):

$$D(\theta) = -\frac{1}{2}\frac{d\lambda}{d\theta}\int_{\theta_n}^{\theta}\lambda d\theta = -\frac{1}{2}\frac{d\lambda}{d\theta}S(\theta_n,\theta) \tag{19}$$

where $\theta_n$ is a reference water content and $S$ is the soil's sorptivity (Philip, 1957b).

A possible functional relationship between $\theta$ and $\lambda$ is the exponential function by Espejo et al. (2014):

$$\theta = \theta_w\left(1 - e^{-c\lambda}\right) \Leftrightarrow \lambda = -\frac{1}{c}\ln\left(1 - \frac{\theta}{\theta_w}\right) \tag{20}$$

where $\theta_w$ is a reference water content different from $\theta_n$ and $c$ is a soil-specific parameter. Integration of equation (20) (right side) from a reference water content $\theta_n$ to a water content $\theta$ allows for a calculation of the sorptivity $S$ (Espejo et al., 2014):

$$\begin{aligned} S(\theta_n,\theta) = -\frac{1}{c}\Bigg[&(\theta_w - \theta_n) \quad \ln\left(1 - \frac{\theta_n}{\theta_w}\right) + \\ &(\theta - \theta_w) \quad \ln\left(1 - \frac{\theta}{\theta_w}\right) + \theta_n - \theta\Bigg] \end{aligned} \tag{21}$$

The reference water content $\theta_w$ is the value towards which equation (20) (left side) will ultimately converge. Therefore, $\theta_w$ was set to the matric saturation water content $\theta_s$ as estimated in section (4.1). The reference water content $\theta_n$ was set to the





assumed residual water content for the mesocosm soils of $0$ resulting in a simplified description of $S$:

$$S(\theta) = -\frac{1}{c}\left[(\theta - \theta_s)\ln\left(1 - \frac{\theta}{\theta_s}\right) - \theta\right] \tag{22}$$

According to equation (19) the diffusivity can then be calculated by combining the first derivative of equation (20) (right side) and equation(22) (Espejo et al., 2014):

$$D(\theta) = -\frac{S(\theta)}{2c(\theta_s - \theta)} \tag{23}$$

The remaining parameter $c$ was estimated by minimizing the sum of squared residuals between values for $\theta$ predicted by equation (20) (left side) and the transformed soil water content data using the optimization algorithm of Nelder and Mead (1965) as implemented in the *optim* function of base R's *stats* package.

### 4.3 Macropore parameter estimation

The two macro flow parameters specific contact length, $L$, and absorption coefficient, $r$, were estimated by inversion of the flow model (Fig. 1). This inversion step prescribed optimal values of the matric flow parameters and the diffusivity function as estimated in sections (4.1) and (4.2) and used those parameters to calculate initial vertical equilibrium profiles. The model simulated the first $12\,\text{h}$ after the start of a precipitation event.

We applied the following objective functions to be minimized:

1. the sum of squared residuals (SSR) between model prediction and measurements of soil water content at depths of $5\,\text{cm}$ and $12\,\text{cm}$ for all sprinkling experiments that exhibited outflow:

$$\sum_{i=1}^{n}\left(\frac{\theta_{i_{obs}}(z) - \theta_{i_{pred}}(z)}{\omega_{i_1}}\right)^2 \tag{24}$$

where $n$ is the number of water content time series, $\theta_{obs}$ and $\theta_{pred}$ are the observed and predicted water contents, respectively, $z$ is soil depth and $\omega_1$ is a weight factor that was set to 10 % of the observed water content.

2. the SSR between the vertical soil water content equilibrium profile $12\,\text{h}$ after the start of the precipitation event as predicted from the matrix parameter optimization and the vertical water content profile predicted by the flow model for the same point in time

$$\sum_{i=1}^{n}\sum_{j=1}^{z}\left(\frac{\theta_{i,j_{obs}}(t=12h) - \theta_{i,j_{pred}}(t=12h)}{\omega_{i,j_2}}\right)^2 \tag{25}$$

where $n$ is the number of vertical profiles, $z$ is the number of simulated layers, $\theta_{obs}$ and $\theta_{pred}$ are the observed and
predicted water contents, respectively, and $\omega_2$ is a weight factor that was also set to 10 % of the observed water content.

The second objective is according to the assumption that flow does not take place later than $12\,\text{h}$ after the start of precipitation and constrained the problem to prevent unreasonable solutions where the soil profile was not in hydrostatic equilibrium.





Optimization was done analogously to matrix parameter optimization by first applying 2500 iterations of the global pseudo-random search optimization algorithm (Price, 1977) as implemented in the *FME* package (Soetaert and Petzoldt, 2010) followed by as many iterations of the (gradient based) quasi-Newton method "BFGS" as implemented in the *optim* function in base R's *stats* package as were necessary to reach convergence (Fig. 2).

## 4.4 Compound uncertainty estimation

In a final step we performed another optimization based on the sum of the first summand in equation (16) and the sum of equations (24) and (25). This optimization used the optimal parameter values estimated in sections (4.1) to (4.3) as starting values.

To estimate parameter uncertainty we then conducted Markov Chain Monte Carlo (MCMC) runs using an adaptive Metropolis algorithm and starting from the final optimized parameter set. Jump lengths were calculated using a multidimensional normal distribution with standard deviations based on the parameter covariance matrix. The covariance matrix was calculated as inverse of the Hessian matrix at the optimized parameter values. The resulting parameter distributions were used to run a model ensemble to estimate prediction uncertainty of vertical equilibrium profiles, water retention functions, diffusivity functions, soil water content time series and seepage fluxes.

## 5 Results

### 5.1 Vertical equilibrium profiles and soil hydraulic properties

Inversions using alternative water retention functions predicted similar vertical equilibrium profiles (Fig. 3). The predicted volumetric water content at the profile bottom differed and was 0.34, 0.35 and 0.32 for the simple exponential, modified exponential and Campbell functions, respectively. Prediction uncertainties of the simple and modified exponential functions did not change much with depth, while the prediction uncertainty of the Campbell function decreased with depth. None of the predicted vertical profiles was within the data range of measurements at 5 cm depth, but the prediction was within 0.02 in all cases. All of the vertical profiles were within the data range of measurements at 12 cm depth. The uncertainties at the profile bottom were reflected by the distributions of $\theta_s$ (Fig. 8).

Within the observed soil water content range the estimated soil water retention functions overlapped and were also in agreement with the water retention functions derived from pedotransfer functions (PTFs) (Fig. 4). All three functions also exhibited a similar slope in the observed water content range, which was lower than the slope predicted by PTFs.

For dry conditions of $\theta < 0.1$, outside the observed water content range, all retention functions predicted much smaller absolute values of matric potential than the PTFs.

The distributions of the estimates of the water retention function parameters reflected their uncertainties (Fig. 5): While the parameter distributions of the parameters $n$ and $\psi_e$ of the Campbell function and the parameters $b$ and $\psi_r$ of the modified





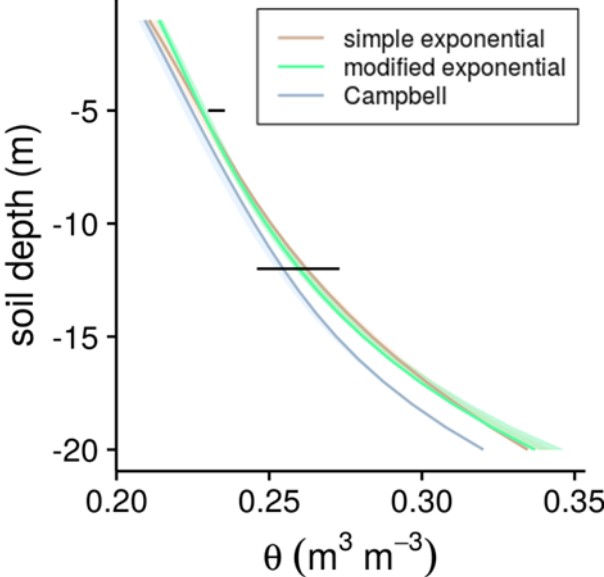

**Figure 3.** Estimated vertical soil water content equilibrium profiles.

Prediction uncertainty (90 % interval) is shown by the shaded areas. The horizontal bars indicate the data range of measurements used in the optimization

exponential function were very narrow, the parameter distributions of the parameters $a$ and $\psi_0$ of the simple exponential function exhibited a much wider spread.

## 5.2 Matric water diffusivity

The estimated matric water diffusivity functions showed a typical shape with a strong decline of diffusivity at low water content values, a strong increase at high water content values and a log-linear increase in the intermediate range (Fig. 6).

Predictions for the different water retention functions were similar. However, there was a difference in the predicted distributions of the diffusion parameter c, which was similar for the simple and modified exponential equations, but shifted by about $2000 \, \mathrm{m}^{-1}\mathrm{s}^{0.5}$ for the Campbell equation (Fig. 8).

## 5.3 Soil water content time series and seepage fluxes

Predicted soil water content time series were very similar among retention functions (Fig. 7). At $5 \, \mathrm{cm}$ depth all predictions were enirely within the range of the measured data. At $12 \, \mathrm{cm}$ depth all predictions showed higher values for a short time frame about $5 \, \mathrm{min}$ after the start of the precipitation, while being witin the range of the measured data for the rest of the time series.





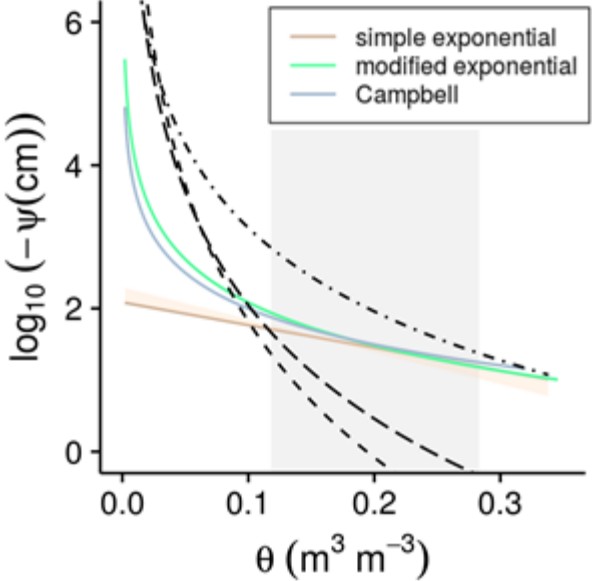

**Figure 4.** Estimated water retention functions.

Prediction uncertainty (90 % interval) is shown by the shaded areas along the solid lines. The other lines indicate the matric potential as calculated using pedotransfer functions by Campbell and Shiozawa (1992) (dashed line), Oosterveld and Chang (1980) (longdashed line) and Saxton et al. (1986) (dotdashed line) (see appendix D for information on pedotransfer functions). The grey rectangle indicates the range of the water content data used in parameter estimation.

All predictions exhibited two consecutive maxima of water content. While both had a similar size at $5 \, \mathrm{cm}$ depth, the earlier maximum was about $0.025 \, \mathrm{m^3 m^{-3}}$ higher than the later maximum at $12 \, \mathrm{cm}$ depth. At both depths the earlier maximum was narrower than the later one.

The distributions of the macropore parameter $L$ and the abstraction parameter $r$ were very similar between the simple and
5   modified exponential functions, while both distributions were shifted towards higher values for the Campbell function (Fig. 8).

With a predicted cumulative outflow of about $250 \, \mathrm{mL}$ the modified exponential function was within the range of the measured cumulative outflow from the mesocosms from $200 \, \mathrm{mL}$ to $300 \, \mathrm{mL}$ (Fig. 9). With a predicted cumulative outflow of $200$ mL the simple exponential function was still at the lower boundary of the data range, while the Campbell function predicted a much higher outflow of about $400 \, \mathrm{mL}$. Prediction uncertainty for the outflow was lowest for the Campbell function and similar
10   for the simple and modified exponential functions.




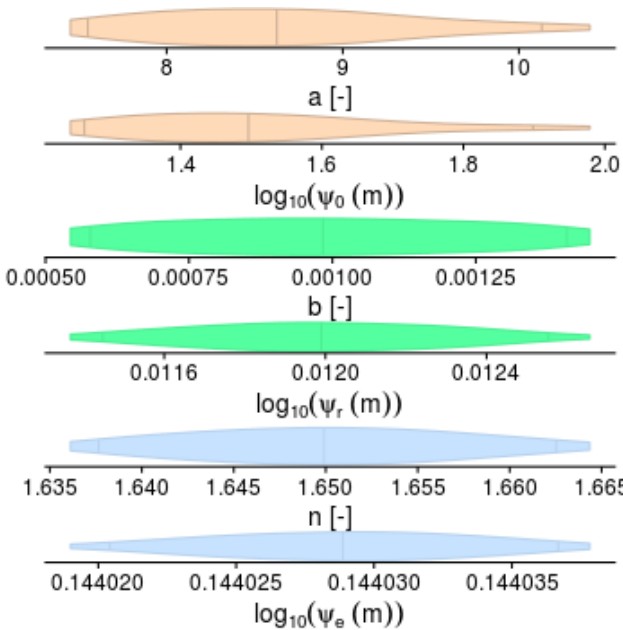

**Figure 5.** Parameter distributions of water retention parameters (90 % intervals). See Table 2 for reference.

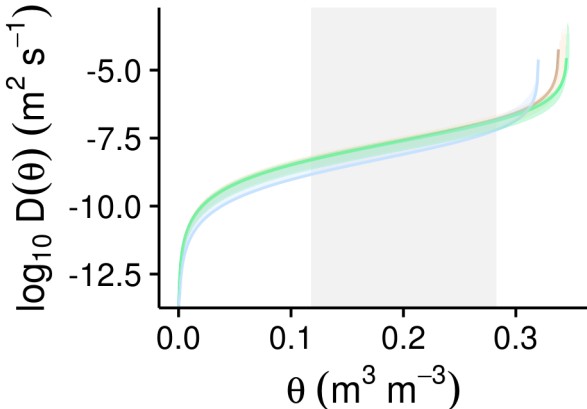

**Figure 6.** Matric water diffusivity functions as predicted by model inversion.

Prediction uncertainty (90 % interval) is shown by the shaded areas along the lines. The grey rectangle indicates the range of the water

content data used in parameter estimation.

## 6   Discussion

The 1D dual permeability model presented in this study has a similar structure to the MACRO model (Jarvis and Larsbo,

2012), but simplified some of its aspects. This was done to decrease the number of required parameters, and to allow a model



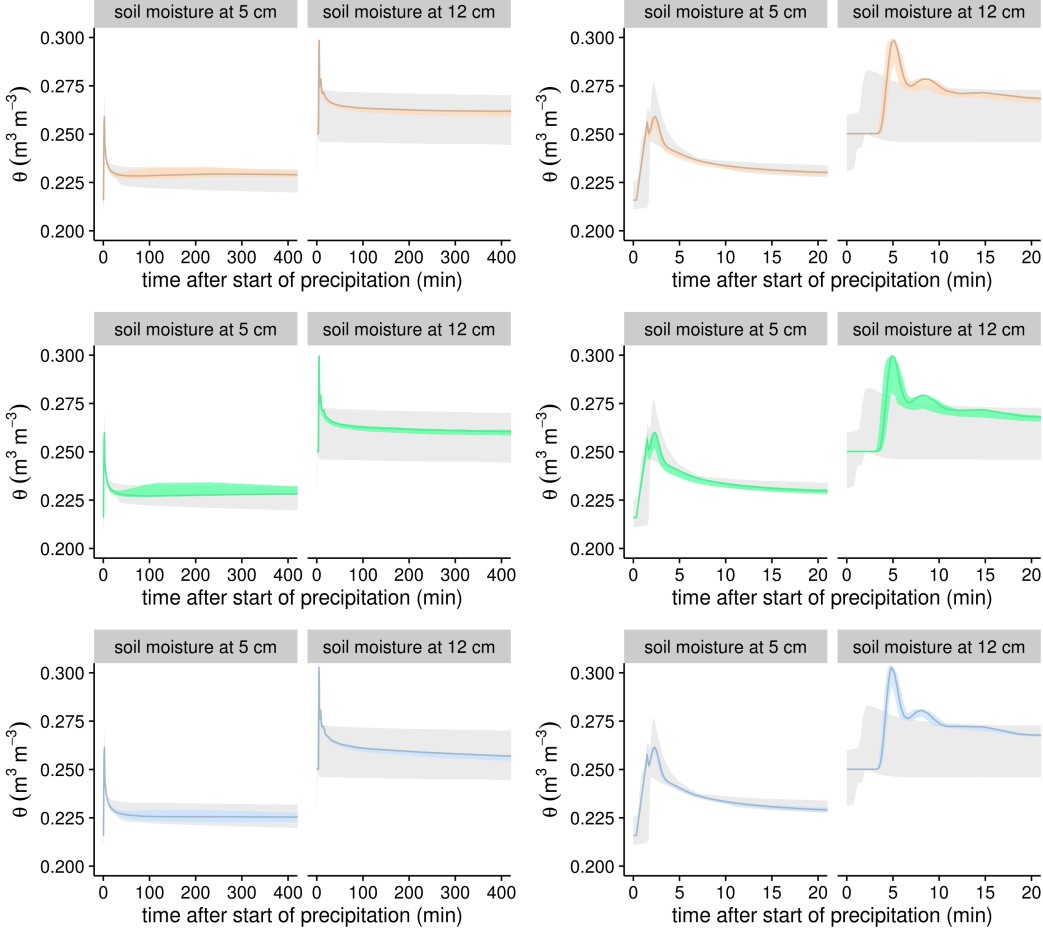

**Figure 7.** Soil water content time series at 5 and 12 cm depth as predicted by the model using simple exponential (top row), modified exponential (center row) and Campbell (bottom row) retention functions. The right column shows the first 20 min of the time series in greater detail.

The coloured shaded areas around the lines show prediction uncertainty (90 % interval). The grey shaded area shows the range of data used in model inversion.

inversion with as few input data as possible with the idea of larger scale application in mind. But it also decreased the model's flexibility and some of the remaining parameters are lumped and do not have a physical meaning anymore.

A novelty of the presented inversion scheme is its (almost) exclusive reliance on soil water content time series data. A major limitation to the use of soil water content is that it may introduce considerable biases, if the soil water retention characteristic is not well known, since water movement in the soil is driven by the potential gradient and not by the gradient in soil water content. This is in addition to other sources of uncertainty in the water retention characteristic like the commonly neglected hysteresis.





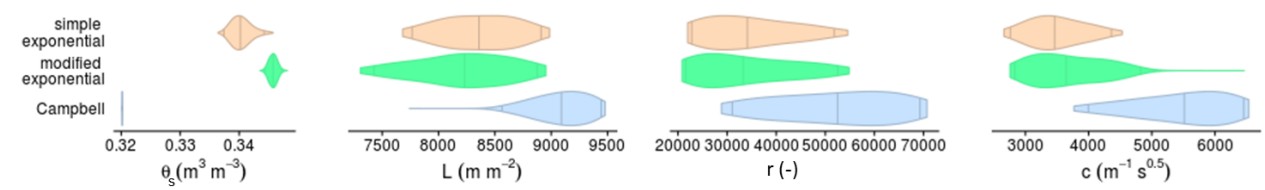

**Figure 8.** Distribution of matric porosity paramter $\theta_s$, macropore flow parameters $L$ and $r$ and diffusivity parameter $c$ as predicted by different retention functions (90 % interval).

This issue could have been alleviated by using soil water potential measurements instead. However, the use of soil water content has the advantage that measurements are more widely available from various sources (eg. the Terrestrial Environmental Observatories (TERENO), the International Co-operative Programme on Assessment and Monitoring of Air Pollution Effects on Forsts (ICP Forests) or the FLUXNET database). In addition soil water content measurement networks with hundreds of

probes are available and allow for the estimation of small scale spatial heterogeneity (e.g. Bogena et al., 2010) and large scale estimates are available from satellite remote sensing.

In general, it is important to stress that the inversion worked well for the particular circumstances in this study with an artificial, homogeneous and shallow soil, but may not be applicable to other conditions. In addition, estimation of macropore parameters requires water content time series that are covering strong precipitation events.

## 6.1 Vertical equilibrium profiles and soil hydraulic properties

The inversion produced meaningful and well restrained vertical soil water content profiles (Fig. 3) and soil water retention functions (Fig. 4). The uncertainty of both, the profiles and the retention functions, decreased considerably between the first and the final optimization for all three retention functions (data not shown for the first optimization), indicating that the simultaneous estimation of all parameters and inclusion of the soil water content time series into the objective function better

constrained the retention function parameters. This approach particularly helped to restrain the matric porosity parameter $\theta_s$ to a very narrow distribution (Fig. 8), which especially decreased the uncertainty of the Campbell retention function (Fig. 4), since it is the only function of the three, that uses $\theta_s$ directly as a parameter (Tab. 4). The higher uncertainty of the predictions by the exponential function displays that it is the least flexible of the three functions in allowing a change in curvature over the course of the profile.

The comparison of the estimated retention functions with PTF predictions shows that all three functions are able to reasonably describe the relationship between soil water content and matric potential in the data range for $\theta$ between 0.1 and 0.3 that was available from the sprinkling experiments (Fig. 4). The underestimation of absolute matric potentials for $\theta < 0.1$ is not surprising given that the functions are merely extrapolated into that water content range. Therefore, using the retention functions to predict water movement in dry soils would result in an overestimation of water flow. A similar problem may arise

for water contents above 0.3 where the predicted absolute matric potentials of all retention functions are at the higher end of the



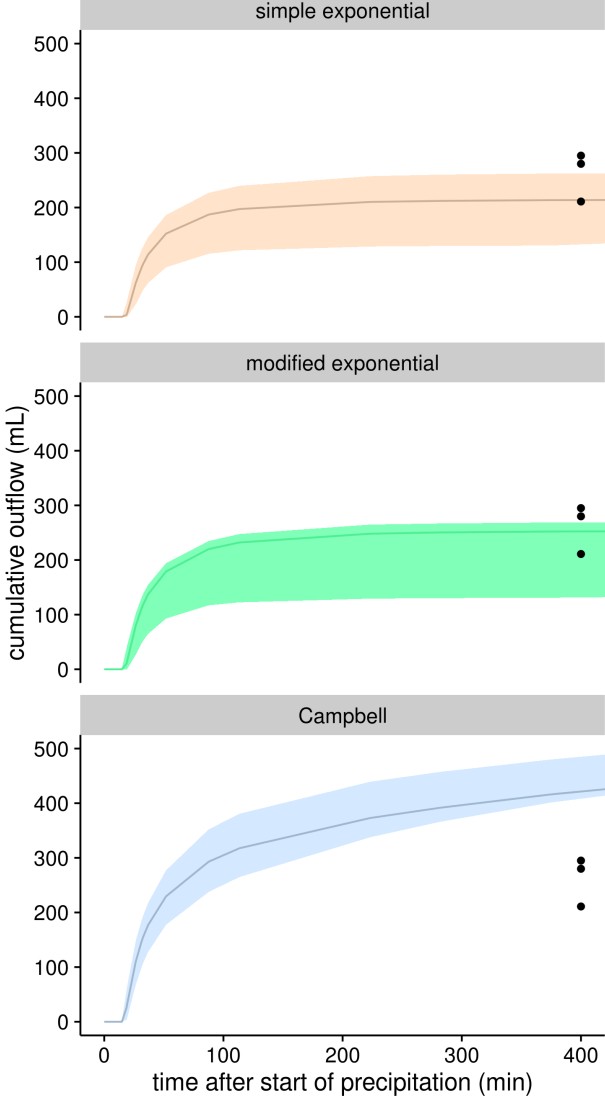

**Figure 9.** Predicted cumulative outflow using simple exponential (top), modified exponential (center) and Campbell (bottom) retention functions.

The points show cumulative outflow as measured from the mesocosms. The shaded areas show the prediction uncertainty (90 % interval).

PTF predictions and therefore the predicted retention functions may underestimate water flow. As expected this is particularly true for the simple exponential function as it deviates massively from the PTF predictions, while the modified exponential and Campbell functions retain a similar shape to the PTFs.

However, neither underestimation of matric flow in wet conditions nor overestimation of matric flow in dry conditions appear
5   to be relevant in the model runs done in this study, since the predicted water content values stayed in the range from 0.2 to 0.3





(Fig. 7). This makes the simple exponential function equally viable to use with this data set, although it would possibly give an implausible result when used with data sets exhibiting a wider range of water contents.

A limited data range and according low credibility of estimated water retention characteristics outside of this range is a common problem when using natural boundary field data for model inversion (e.g. Over et al., 2015; Scharnagl et al., 2011; Wollschläger et al., 2009), and could not be avoided in the mesocosms as well, since more extreme soil moisture conditions could have damaged the grasses. If a sufficiently long time series is available an estimate of the effective residual water content may be obtained by using the lowest measured value. In the wet range the largest measured water contents are already close to the predicted matric porosity suggesting that problems with extrapolation into this water content range should be minor.

An important limitation to the vertical equilibrium approach is that vertical water content profiles can be discontinous in layered soil profiles (Wollschläger et al., 2009) and therefore a meaningful interpolation of water content measurements may be impossible to obtain. This could be circumvented by piecewise interpolation or use of a step function, if at least one water content time series is available for every (hydrologically) different layer. We do not expect this to play a role in the mesocosm soils since they were quite homogeneous due to their setup and shallow depth of 20 cm.

## 6.2   Matric water diffusivity

Application of the method of Espejo et al. (2014) to the mesocosm data resulted in useful estimates of the soil water diffusivity function (Fig. 6). Similar to the retention functions their applicability was limited by the range of available water content data. Their uncertainty could be further reduced by using longer drying periods resulting in lower water contents. The different retention functions hardly affected resulting diffusivity functions. The predicted diffusivity and retention functions also allowed for calculation of the water conductivity function for the matrix domain (Espejo et al., 2014) (data not shown).

## 6.3   Soil water content time series and seepage fluxes

The comparison between predicted soil water content time series and range of measurements (Fig. 7) shows that the model is able to reproduce the essential features of the time series. However, the wide distributions of the macropore flow parameters $L$ and $r$ (Fig. 8) indicate that the macropore model is far less constrained by the soil water content data than the matric flow model. This is hardly surprising given the much larger number of water content time series available to constrain the matrix domain parameters compared to the single event with seepage flow used to constrain the macro flow parameters.

Since the outflow in the mesocosm experiment is only composed of macropore flow the high uncertainty of the macropore flow parameters also results in high uncertainty of outflow predictions (Fig. 9). The strong influence of the macropore flow parameters on outflow prediction is also reflected in the parameter distributions (Fig. 8). The simple and modified exponential functions exhibit similar distributions of $L$ and $r$ and predict a similar outflow, while the Campbell function differs in both regards (Fig. 9).

The soil water content time series predicted by the model shows two peaks at both depths that in the measurements are only indicated at 5 cm depth and do not occur in the measured time series at 12 cm depth. This suggests that the peaks either superimpose each other or either the matric flow peak or the macropore flow peak is so small that its amplitude is below the





measurement resolution of the soil water content probes ($0.001\,\mathrm{m^3 m^{-3}}$). Indeed, the model predicts a combination of both: At $5\,\mathrm{cm}$ depth both peaks are so close to each other that they almost superimpose while the predicted matric flow peak has an amplitude of only $0.004\,\mathrm{m^3 m^{-3}}$ at $12\,\mathrm{cm}$ depth. (Fig. 7). Accordingly, it is possible that the second peak (the matric flow peak) is not seen in the aggregated measurements. The alternative that the macropore flow peak is not seen in the measurements is unlikely because it is contrasted by the indicated two peaks at $5\,\mathrm{cm}$ depth and the strong rooting observed in the mesocosms, which is likely to have created preferential flow paths.

The prediction of the water content time series at $12\,\mathrm{cm}$ depth is somewhat worse than the prediction at $5\,\mathrm{cm}$ and there is no indication how the model would perform at greater depths. Therefore, we plan to apply the model to field data from a mixed beech forest plot in the AquaDiva critical zone observatory (Küsel et al., 2016). The field soil will be deeper and more heterogeneous than the substrate in the mesocosms.

## 7 Conclusions

In this study we presented a simplified 1D dual permeability model whose structure is similar to the MACRO model (Jarvis and Larsbo, 2012). The model parameters were estimated from soil water content time series data in a novel inversion scheme. The study shows that a meaningful soil water content based calibration of a 1D dual permeability model is possible at least under the specific circumstances of an artificial, relatively homogeneous and shallow soil, if the model and water retention functions are kept simply enough. The presented model and inversion method provide means to exploit soil water content measurements of local sensor networks like the *SoilNet* (e.g. Bogena et al., 2010) without the need to acquire additional data. Hence, this study is a critical step towards estimating seepage fluxes at sites with available soil water content time series data. Furthermore, the presented model calibration will allow exploring stand-scale heterogeneity in water fluxes at sites equipped with soil water content sensor networks.

## Appendix A: Detailed derivation of the macropore flow model

This section mainly follows the work of Hincapié and Germann (2009).

In order to keep the moving water film intact viscous shear forces have to balance out gravity at every point in the water film, so:

$$mg = \tau A_c \tag{A1}$$

where $m$ is the mass of water in the water film, $g = 9.81\,\mathrm{m\,s^{-2}}$ is the gravitational acceleration on earth and $\tau$ is the shear stress induced on the contact area between the water film and the macropore walls ($A_c$) by the moving water.

Shear stress in the water film moving in parallel to the macropore surface can be written in terms of either dynamic or kinematic viscosity:

$$\tau(f) = \mu \frac{\partial v}{\partial f} = \eta \rho \frac{\partial v}{\partial f} \tag{A2}$$





where $f$ is the height above the interface between the water film and the pore wall, $\mu$ is the dynamic viscosity of water, $v$ is the flow velocity of a water lamina at distance $f$ from the interface, $\eta$ is the kinematic viscosity of water and $\rho$ is the density of water.

The water mass at a distance $f$ above the macropore wall, that has to be balanced by the shear forces is that of the water that is in greater distance from the interface. It can be calculated as:

$$m = (F - f)lz\rho \tag{A3}$$

where $F$ is the overall film thickness, $l$ is the contact length between water film and pore wall and $z$ is the vertical length of the water film.

Combining A1 to A3 leads to:

$$(F - f)zlg\rho = \frac{\partial v}{\partial f}\eta\rho lz$$
$$\frac{\partial v}{\partial f} = (F - f)\frac{g}{\eta} \tag{A4}$$

Integration from 0 (pore wall interface) to $f$ gives the velocity of the lamina at a distance $f$ from the water–solid interface:

$$v(f) = \int_0^f \frac{dv}{df}df = \int_0^f (F - f)\frac{g}{h}df$$
$$v(f) = \frac{g}{h}f\left(F - \frac{f}{2}\right) \tag{A5}$$

The differential volume flux of a film lamina at a distance $f$ from the pore wall interface is the product of the lamina water mass and the velocity. The differential volume flux density can be calculated by dividing the volume flux by the macroscopic cross sectional area:

$$dq(f) = \frac{1}{A}v(f)ldf \tag{A6}$$

where $q$ is the volumetric flux density and $A$ is the macroscopic cross sectional area.

The volume flux density $q$ is also the product of an average velocity of the downward moving water front $v_W$ and the macro pore water content $W$:

$$q = v_W W \tag{A7}$$

The film thickness can also be expressed in terms of the macro pore water content:

$$F = \frac{W}{L} \tag{A8}$$

The average velocity of the downward moving water can then be expressed by combining A7 to A8:

$$v_W = \frac{1}{W}\frac{1}{3}\frac{g}{\eta}\frac{W^3}{L^3}L = \frac{g}{3\eta}\left(\frac{W}{L}\right)^2 \tag{A9}$$





This velocity can now be used in an advection equation to receive the equation governing the movement of the macropore water:

$$\frac{\partial W}{\partial t} = -v_W \frac{\partial W}{\partial z}$$
$$\frac{\partial W}{\partial t} = -\frac{g}{3\eta} \left(\frac{W}{L}\right)^2 \frac{\partial W}{\partial z} \tag{A10}$$

## Appendix B: Derivation of vertical equilibrium profile function for modified exponential function

Inserting the modified exponential function (R2) into equation 15 and rearranging gives:

$$\frac{e^{-b\theta}}{\theta^2} = \frac{e^{-b\theta(z_{ref})}}{\theta(z_{ref})^2} + \frac{z_{ref} - z}{c} \tag{B1}$$

Substituting the right hand side with $y$ and further rearrangement leads to:

$$\theta^2 e^{b\theta} = \frac{1}{y}$$
$$\frac{b\theta}{2} e^{\frac{b\theta}{2}} = \frac{b}{2}\sqrt{\frac{1}{y}} \tag{B2}$$

Now substituting $t = \frac{b\theta}{2}$ results in:

$$t e^t = \frac{b}{2}\sqrt{\frac{1}{y}} \tag{B3}$$

Equation B3 can be solved for $t$ using the definition of the Lambert-W function (omega function, product logarithm):

$$t = \omega\left(\frac{b}{2}\sqrt{\frac{1}{y}}\right) \tag{B4}$$

Resubstituting and rearranging results in the equation for the vertical equilibrium profile (R2):

$$\theta(z) = \frac{2}{b}\omega\left(\frac{b}{2}\sqrt{\frac{1}{y}}\right)$$
$$y = \frac{e^{-b\theta(z_{ref})}}{\theta(z_{ref})^2} + \frac{z_{ref} - z}{c} \tag{B5}$$

## Appendix C: Aggregation and ET correction of soil water content time series data

While the high temporal resolution of the soil water content time series data (10 sec) was required to capture the rapid dynamics right after the precipitation events, it did result in a large data stream for the remainder of the time series that did not contain much information, but was quite costly to work with. Therefore, the data were aggregated to yield new time series with fewer measurements in their less dynamic parts.

For aggregation the time series were divided into a monotonically rising and a monotonically declining limb. All higher frequency fluctuations in water content apart from that were considered measurement noise. Due to the limited measurement



resolution of the soil water content probes of $0.001$ m$^3$ m$^{-3}$ there was only a limited number of discrete values left in each limb resulting in a step-wise time series. Assuming monotony the mean of the first and last occurrence times of each discrete water content step were used as the time of a single observation of the respective soil water content. The mean positions then were assigned the according values. This procedure maintains a high (temporal) density of measurements in parts of the time

series that exhibit high dynamics, while drastically reducing the amount of data in less dynamic parts transforming the discrete steps into a continous time series. The aggregation results in a relatively stronger emphasis on dynamic parts of the time series during inverse parameter estimation.

Evapotranspiration (ET) is not modelled explicitly by the model. Instead, a fixed ET rate is prescribed. The use of a constant ET rate appears to be reasonable since the mesocosms were kept in a climate chamber under constant climatic conditions.

Variation of ET introduced by the day and night cycle applied to the mesocosms will be periodic and even out in a linear interpolation. In order to estimate ET rate the falling limb of the time series was partitioned into an exponentially declining and a linear declining part by obtaining the best combined fit of an exponential and a linear function to the aggregated data. It was assumed that the exponentially declining part of the falling limb is still affected by rapid vertical water movement, while the linear part is related to a much slower process, which is ET. Therefore, we assume that the constant rate of ET equals the

slope of the linear fit.

## Appendix D: Pedotransfer functions

The water retention functions as estimated by model inversion were compared with three water retention functions by Campbell and Shiozawa (1992), Saxton et al. (1986) and Oosterveld and Chang (1980). The formulations of these functions were taken from the Appendix of Guber et al. (2006). The three PTFs use soil texture data to predict the parameters of the Brooks-Corey

equation (Brooks and Corey, 1964) and were chosen specifically, since they assume a residual water content content of zero and therefore treat the special case of the Brooks-Corey function that is the Campbell function. Measured particle size classes had to be converted from the German size class system to the USDA system in order to use the PTFs. This only affected the relative proportion of silt and sand particles. Conversion was done by loglinear interpolation (Nemes et al., 1999), which appeared to be the most viable method with data availabe only for three size classes.

*Acknowledgements.* We thank Nicholas Jarvis for helpful hints and discussions and Markus Reichstein for helpful discussions and comments on the manuscript.

The work has been funded by the Deutsche Forschungsgemeinschaft (DFG) CRC 1076 "AquaDiva". Field work permits were issued by the responsible state environmental offices of Thüringen. We thank the Hainich CZE site manager Robert Lehmann, Christine Hess for scientific coordination and the Hainich National Park.

Climate chambers to conduct experiments under controlled temperature conditions were financially supported by the Thüringer Ministerium für Wirtschaft, Wissenschaft und Digitale Gesellschaft (TMWWDG; project B 715-09075).





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



**Table 1.** List of symbols

| Symbol | Description | Unit |
|---|---|---|
| **Macropore domain** | | |
| *State variables* | | |
| $W$ | macropore water content | $\mathrm{m^3 m^{-3}}$ |
| $F$ | thickness of macropore water film | m |
| $l$ | contact length between water film and macropore wall | m |
| *Properties* | | |
| $A_c$ | cross sectional area | $\mathrm{m^2}$ |
| $S_{ma}$ | effective saturation of macropore domain | $\mathrm{m^3 m^{-3}}$ |
| *Fluxes* | | |
| $q_{ma}$ | volumetric flux in macropore domain | $\mathrm{m^3 m^{-2} s^{-1}}$ |
| $q_{top_{ma}}$ | rate of infiltration into macropore domain | $\mathrm{m^3 m^{-2} s^{-1}}$ |
| $v_W$ | advection velocity of macropore water film | $\mathrm{m\ s^{-1}}$ |
| **Matrix domain** | | |
| *State variables* | | |
| $\theta$ | matric water content | $\mathrm{m^3 m^{-3}}$ |
| $\theta_1$ | water content of topmost grid cell | $\mathrm{m^3 m^{-3}}$ |
| $\theta_n$ | lower reference water content for diffusivity function | $\mathrm{m^3 m^{-3}}$ |
| $\theta_r$ | residual water content | $\mathrm{m^3 m^{-3}}$ |
| $\theta_w$ | upper reference water content for diffusivity function | $\mathrm{m^3 m^{-3}}$ |
| $\theta_{z_{ref}}$ | water content at a reference depth | $\mathrm{m^3 m^{-3}}$ |
| $\psi$ | matric potential | m water column |
| *Properties* | | |
| $D$ | matric water diffusivity | $\mathrm{m^2 s^{-1}}$ |
| $K$ | matric hydraulic conductivity | $\mathrm{m\ s^{-1}}$ |
| $K_{top}$ | hydraulic conductivity of topmost matrix layer | $\mathrm{m\ s^{-1}}$ |
| $S$ | matric sorptivity | $\mathrm{m\ s^{-0.5}}$ |
| *Fluxes* | | |
| $q_{mi}$ | volumetric flux of matrix domain | $\mathrm{m^3 m^{-2} s^{-1}}$ |
| $q_{top_{mi}}$ | rate of infiltration into matrix domain | $\mathrm{m^3 m^{-2} s^{-1}}$ |
| **Other** | | |
| $t$ | time | s |
| $z$ | soil depth (positive downwards) | m |
| $z_{ref}$ | reference depth | m |
| $\lambda$ | Boltzmann variable | $\mathrm{m\ s^{-0.5}}$ |
| *Fluxes* | | |
| $A$ | abstraction from macropore water film into matrix domain | $\mathrm{m^3 m^{-3} s^{-1}}$ |
| $ET$ | evapotranspiration | $\mathrm{m^3 m^{-3} s^{-1}}$ |
| $P$ | precipitation rate | $\mathrm{m^3 m^{-2} s^{-1}}$ |
| *natural constants* | | |
| $g$ | gravitational acceleration | $\mathrm{m\ \ s^{-2}}$ |
| $\eta$ | kinematic viscosity of water | $\mathrm{m^2 s^{-1}}$ |





**Table 2.** List of model parameters

| Parameter | Symbol | Unit |
|---|---|---|
| **Matrix flow parameters** | | |
| *Simple exponential approximation* | | |
| shape parameter | $a$ | - |
| matric potential at 0 water content | $\psi_0$ | m |
| matric water content of bottom grid cell | $\theta_{z_{ref}}$ | $\mathrm{m^3 m^{-3}}$ |
| *Modified exponential approximation* | | |
| shape parameter | $b$ | - |
| reference matric potential | $\psi_r$ | m |
| matric water content of bottom grid cell | $\theta_{z_{ref}}$ | $\mathrm{m^3 m^{-3}}$ |
| *Campbell (1974)* | | |
| shape parameter | $n$ | - |
| air entry water potential | $\psi_e$ | m |
| matric saturation water content | $\theta_s (= \theta_{z_{ref}})$ | $\mathrm{m^3 m^{-3}}$ |
| **Diffusivity parameters** | | |
| diffusivity function shape parameter | $c$ | $\mathrm{m^{-1} s^{0.5}}$ |
| (upper) reference water content | $\theta_w (= \theta_{z_{ref}})$ | $\mathrm{m^3 m^{-3}}$ |
| (lower) reference water content | $\theta_n (= 0)$ | $\mathrm{m^3 m^{-3}}$ |
| **Macro flow parameters** | | |
| specific contact length | $L$ | $\mathrm{m^1 m^{-2}}$ |
| abstraction coefficient | $r$ | - |





**Table 3.** Optimization workflow and data usage in each optimization step

| | Optimization step | Data used |
|---|---|---|
| 1. | matric retention parameters | volumetric soil water content prior to precipitation and at 12 h after precipitation and mesocosm water balance using data only from events exhibiting seepage. |
| 2. | matric diffusivity parameters | volumetric soil water content starting more than 12 hours after precipitation using data from all measured events |
| 3. | macropore parameters | volumetric soil water content time series up to 12 hours after precipitation using data only from events exhibiting seepage |
| 4. | final inversion | volumetric soil water content prior to precipitation and at 12 h after precipitation, soil water content up to 12 hours after precipitation using data only from events exhibiting seepage |

**Table 4.** Water retention functions, their derivatives and according vertical equilibrium profile functions

| Name | Function $\psi(\theta)$ | First derivative $\frac{d\psi}{d\theta}$ | Vertical equilibrium profile $\theta(z)$ | parameters | |
|---|---|---|---|---|---|
| Simple exponential approximation | $\psi_0 e^{-a\theta}$ | $-a\psi_0 e^{-a\theta}$ | $-\frac{1}{a}ln\left[e^{-a\theta(z_{ref})} + \frac{z_{ref}-z}{|\psi_0|}\right]$ | $a, \psi_0$ | (R1) |
| Modified exponential approximation [1] | $\frac{1}{\theta^2}\psi_r e^{-b\theta}$ | $-\frac{\psi_r(b\theta+2)e^{-b\theta}}{\theta^3}$ | $\frac{2}{b}\omega\left(\frac{b}{2}\sqrt{\frac{1}{y}}\right)$ $y = \frac{e^{-b\theta(z_{ref})}}{\theta(z_{ref})^2} + \frac{z_{ref}-z}{|\psi_r|}$ | $b, \psi_r$ | (R2) |
| Campbell (1974) | $\psi_e\left(\frac{\theta}{\theta_s}\right)^{-n}$ | $-\frac{n\psi_e}{\theta}\left(\frac{\theta}{\theta_s}\right)^{-n}$ | $\theta_s\left[\left(\frac{\theta(z_{ref})}{\theta_s}\right)^{-n} + \frac{z_{ref}-z}{|\psi_e|}\right]^{-\frac{1}{n}}$ | $n, \psi_e, \theta_s$ | (R3) |

[1]: $\omega(\ )$ is the Lambert-W function (omega function, product logarithm). Since $y > 0$ and $a > 0$ the term in the brackets is also always larger than 0.
Therefore, any calculations in the model only require the principal branch of the Lambert-W function, which is defined on the interval $[-\frac{1}{e}, \infty)$. The model
uses an implementation of the Lambert-W function provided by the R-package lamW (Adler, 2016).