# Peer review of "Modeling macropore seepage fluxes from soil water content time series by inversion of a dual permeability model"

_Hydrology and Earth System Sciences, 2017_

## Referee Comment (RC1) · Anonymous Referee #1 · 22 Jun 2017

The idea presented in this paper is good, but the execution is seriously, possibly fatally, flawed. The rationale for a simple, parsimonious model that can be calibrated using only data that can be obtained with relative ease is convincingly presented. The combination of a simplified macropore model with a matric flow model based on retention and diffusivity curves with few parameters is worth exploring.

But then the difficulties start:

An experiment was carried out in which a soil with low clay content was sieved, homogenized and mixed with fine gravel. If one wished to avoid the development of a macropore network, this was the way to go. Grass was sown, which was well watered.

[Figure]

Late in the paper it is stated that the roots were expected to create macropores, but they occupied these themselves. Dye tracer tests to verify after the experiment if there was preferential flow were not performed. The soil was kept moist so the chances of root shrinkage (opening up an air gap between the root and its macropore wall) were small. The shallow depth of the soil (20 cm) and the lower boundary with only a single outlet that presumably only permitted water to flow out at atmospheric pressure also ensured wet conditions throughout. Thus, the experiment allowed only a limited range of water content to be examined under the absence of macropores.

The macropore model has its own problems:

The expression for the flow velocity in the macropores is a factor 3 off. I presented an alternative derivation near Eq. (2). Eq. (4) is incorrect as well: the flux term in the mass balance is formulated incorrectly. I presented an improved version. The resulting version of the flow equation (the second part of Eq. (4)) is the same though. It appears that the error in the flow velocity and the mass balance term canceled out, perhaps that is why they went unnoticed.

The soil physics is also questionable.

The conductivity at the top soil is falsely interpreted as the maximum infiltration rate (section 2.4). It is better to state that the maximum infiltration rate is determined by the hydraulic gradient and the hydraulic conductivity at the tip of the wetting front. That is why the gradients in the wetted part of the soil during infiltration are often not very large. The wetted soil is so conductive compared to the dry soil at the tip of the wetting front that hardly any gradient is needed to transport the infiltrated water to the tip of the wetting front. Therefore, the conductivity at the soil surface is only important during the initial stage of wetting.

This is illustrated by the fact that that the infiltration capacity becomes lower as the wetting front moves down. At the start, the top fraction of a mm of the soil is saturated, and the gradient there can be in the order of 10000. As infiltration proceeds, the front

becomes less sharp, and the flow through the wetted part of the soil causes a (relatively minor) drop in the matric potential as the water travels towards the wetting front. Both phenomena (the former often much more than the latter) lead to a reduction in the hydraulic gradient at the wetting front. With the soil at the tip of the wetting front being equally dry, this leads to a diminished infiltration capacity. When the wetting front arrives in a wetter part of the soil, the local conductivity increases a lot, and the infiltration capacity might well increase then.

All in all, approaching matric infiltration based only on the conductivity at the soil surface, and assuming that is somehow a maximum flow rate is not very good. There are several infiltration equations (by Green-Ampt, Philip, and others) that should work better, especially in the early stages (e.g., Hillel, 1998, pp. 391-391).

In section 4.1, unit gradient conditions are confused with hydrostatic equilibrium. The authors claim that d theta / dz = 0 indicates hydrostatic equilibrium. But this is the expression for unit gradient flow, not that for hydrostatic equilibrium.

Unit gradient flow: flow driven by gravity only; d psi/dz = 0, and therefore d theta/dz = 0.

Hydrostatic equilibrium: hydraulic gradient = 0: dH/dz = 0. This implies that the gradient in the matric potential cancels out the gradient in the gravitational potential. For z positive downward, as it is defined here: d psi/dz = 1. Therefore, d theta /dz <> 0.

The soil moisture profile can be derived from the soil water retention curve if the plane where psi = 0 (phreatic level) is known. The authors apply this technique later in the paper.

The casual statement in section 4 that ET had no effect even though a 20 cm closed container apparently was not at hydrostatic equilibrium (contrary to what the authors believe) also betrays a lack of understanding. Root water uptake tends to make soil moisture profiles in non-layered soils more uniform because the roots take up more

water where it is easier to get.

The explanation of Eq. (14) is simply false (and its sign is wrong). The expression for the matric potential follows directly from the definition of hydrostatic equilibrium (zero hydraulic gradient). Applying Eq. (8) for steady-state conditions is less specific (constant but non-zero flows are also steady state), and taking the derivative of the equation leads to additional terms that were conveniently ignored. In addition, the sign of the equation is only correct if z is defined positive upward, which contradicts Table 1.

In Equation 15, the sign is wrong (a consequence from the error in Eq. 14). Apparently it escaped the attention of the authors that they calculate positive matric potentials in unsaturated soils with this equation.

The exponential retention curves in Table 4 behave non-physically. The exponential terms in them go to 1 as theta goes to zero and to some small but finite value as theta goes to theta-s. If the exponent is made proportional to (theta-s minus theta) instead of theta they behave as they should.

In section 4, the Boltzmann transformation is applied (Eq. 19). This approach is only useful when capillarity dominates gravity (thereby ensuring the validity of Eq. 18, where the gravity term is omitted) and conditions far away from the infiltration front have no effect because the transformed equation is solved for a semi-infinite column with a prescribed water content at infinity.

The authors waited for 12 hours after rainfall and had only 20 cm long columns. The long wait diminished the matric potential gradients, thus ensuring that the gravitational gradient was no longer negligible. Also, the infiltration front had penetrated most if not all of the column by then, invalidating the semi-infinite approximation. The resulting values of the diffusivity are therefore meaningless. The fitted diffusivity parameters had an effect on the other parameters when in the last leg of the optimization process, all parameters were fitted simultaneously, but it is impossible to determine how detrimental they were. In any case, they cast doubt upon all fitted parameter values and the model

runs based on these values.

Given that the paper suffers from both experimental and theoretical flaws I doubt if a revision can save it.

Hillel, D. 1998. Environmental soil physics. Academic Press, San Diego, U.S.A.

Please also note the supplement to this comment: http://www.hydrol-earth-syst-sci-discuss.net/hess-2017-336/hess-2017-336-RC1-supplement.pdf

[Figure]

**Supplement:**

[revised manuscript text omitted]

---

## Referee Comment (RC2) · Anonymous Referee #2 · 17 Jul 2017

The paper presents a dual permeability (DPM) simplified 1D model by combining a Richard's type of solution for the matrix flow and a Kinematic wave solution for the macropore domain. The simplification allows to reduce the number of parameters which allows an easier implementation of an inverse modeling subroutine. The model was calibrated and the uncertainty of input and output parameters were both estimated with the FME package for R.

The reduction idea plus the combination of mathematical 1D alternative formulation of DPM, validation with lab experimental measurements and the quantification of input and output uncertainty makes a complete approach to the problem.

[Figure]

The model simplification and input reduction is always desired for models as their over-parametrization makes their implementation difficult and it may even increase their uncertainty. Therefore, the intention of reducing the number of parameters by simplifying the model from a mathematical point of view is the strong selling point of this study.

However, the study points out that their aim is divided in two main parts; Developing the low complexity model and develop a model inversion scheme. For the former one, it is not clear to the reader or reviewer if lower complexity model is desirable as even the authors point out that the model can only be implemented for the specific case of the controlled conditions in the lab, the obtained results are not very good, they are only tested for one infiltration event of the 15 intended and last but not least, it reduces the flexibility for fitting the capacity of retention functions.

For the latter part, it seems that the necessity of the inversion modeling process for the calibration is not sufficiently explained and seems merely a product of the availability of a software package. This last comment doesn't mean it cannot be done but it shouldn't be one of the main goals of the study as all models in any field require calibration and validation. If such a robust implementation is really required, the study should also probe it by comparing the results of different calibration methods so that the expected drawbacks (equability and local minima) are proven to exist.

If performing all these additions and addressing the comments for improving the paper, I will accept the paper with major changes.

Regarding the model development section:

Please explicitly include the reduction of parameters (which parameters are required if no simplification is done (presented Table 2) vs which parameter are required in the proposed model) with respect to the different methods so that the study becomes more attractive to the reader.

It is important to clarify that the matric water diffusivity referred in the study is a "borrowed" term to refer to describe the gradient in time of the soil water content (in the differential form of a diffusion process) instead of a diffusive particle solute movement in gases or liquids. Hence it is recommended to use the term Hydraulic diffusivity as explained for example in the book of Hillel "Introduction to environmental Soil Physics".

The deduction of the macropore flux is missing a scheme (i.e. as presented by Hincapié and Germann 2009) in which its deduction and variables are presented for the new proposed model representation.

The exchange lumped term r is a function the same 3 parameters (G,d and ) which are present in both Dual permeability and dual porosity models described in literature (See Gerke and Van Genuchten 1993, Evaluation of a first order water transfer term for variably saturated dual-porosity flow models). It would be interesting to compare the obtained results with the ones presented in literature as the posterior r distribution seems to reflect a high degree of uncertainty in the model. Furthermore, it would be valuable if this macropore exchange parameters where fixed based on literature and see how much they either reduce or increase the uncertainty bounds which will allow to have a first impression of their importance for the studied soil in the proposed model.

According to the manuscript, 45 time series extracted from 15 rainfall events were measured but only 1 of them presented observable seepage fluxes which might imply that the experiment was not correctly designed. It might be related to the high silt content of the soil probes. Also note that the method presented in Campbel (1974) was validated against sandy loam (higher content of sand than silt).

Regarding the inverse modeling scheme:

It is not explicitly mentioned which are the input parameters which are going to be found by the inverse modeling algorithm for each of the optimization steps.

A very "robust" algorithm for inverse modeling is used for fitting a rather simple model (as pointed by the authors) with a reduced number of input parameters which gives the

impression that its implementation might not be required. Two assumptions are pointed to support the use of the algorithm (e.g existence of local minimums and equifinality. For the former one, it must be proved that this is the case by implementing a simpler algorithm in which local minima is obtained. It is expected that with monotonous and continuous functions, the chances of finding local minima for the objective functions presented are reduced. For the latter one, it is expected that this is still the case as too many parameters (e.g. for the first inversion a„b, n and L,r,c and theta_s) are fitted from basically two measurements of soil water content (5 cm and 12 cm). All water retention functions are not linear which implies that at least 3 different measurements should be used for a reliable fitting

The uncertainty weights (w1 and w2) used inside the objective functions for minimization are chosen arbitrarily. Please explain the choices or even if possible, make a small sensitivity test as they can bias the obtained distributions for the unknown parameters.

The authors imply that a double optimization inverse modeling framework is needed to avoid equifinality. So the questions that arise are: why are the authors want to avoid equifinality while including more than one water retention function as part of the study (Note that equifinality also implies that different models may perform similarly) and is it important to avoid it when the parameter results are presented as uncertainties for each parameter?

It will be very valuable to compare parameters obtained from the initial inverse modeling with respect to the same ones obtained in the next inverse modeling. Further more, if one of the main goals of the manuscript is to show the importance of the whole inversion method (Besides the simplification of the model as pointed in page 3, line 20), a comparison against a single inverse modeling optimization of all parameters will benefit the study significantly. Otherwise, what is the point of doing 2 step optimization without probing the effects of equifinality or local minima encountering ?

As a suggestion: For this kind of studies, the GLUE (Beven and Binley) methodology

may be a better option as it accepts the fact that equifinality is present for all paramters and models and hence evaluates the models based on a likelihood function.

Results and discussion:

In figure 3, the range of measurements differs from the ones obtained for the estimated functions. If these curves were used as seed values for the next optimization, would it imply that the measurements obtained at -5 cm are useless? Otherwise explain the implication of not having the fitted functions inside the measured bounds. Moreover, is it necessary to perform the initial model inversion? Ideally this step will avoid the entrapment on local minima but evidence of such value must be supported, especially for the fitting on monotonic functions such as the ones presented in this manuscript.

In figures 3 and 4, it is not clear how the 90% confidence intervals where obtained and they are almost non-visible, especially in figure 4.

In Figure 4, it is also not clear why the dashed line (Shiozawa) and long dashed line (Oosterveld and Chang) cut the horizontal axis between 0.2 and 0.3 theta values. Whereas the predicted values may go as far as 1.0.

In section 6.1 it is stated that there was a significant reduction of uncertainty after the second model inversion step but it is also acknowledged that data from the first optimization is not shown. Why? What would be the result with one general inversion modeling of all the parameters at the same time?

In the last paragraph of the conclusions it is stated that the prediction of the water content time series at 12 cm is somewhat worse than the prediction at 5 cm. However, the parameter estimation is better fitted for the measurements at 12 cm than the ones obtained at 5 cm which also are said to be close to the global optimum (see Figure 3). But appreciations would mean that the author is either getting good results at 5 cm with worst fitting values at this depth and is getting bad results in the moisture content time series at 12 cm while having a good fit of the retention function at that depth. This

may be a good evidence that one inverse modeling procedure of all parameters at the same time might be better (but more complicated) than making a first optimization for the water retention functions as a not so "optimal" and restrictive

Soil heterogeneity and uncertainty derived from the initial conditions and its implications in the model is not further discussed in the manuscript.

Minor comments:

It is strongly suggested to make a more detailed flow chart of the whole inverse modeling process in a flow chart in which the input and output is clearly showed as figure 2 tells very little information.

Explain clearly the difference between calibration and inverse modeling (the first a general concept and the second one a method for calibration and estimation of input uncertainty) as both terms are used indiscriminately in the abstract, section 3 and conclusions.

In section 4, the expression "A first inversion. . ." is used but the expression "The second inversion . . ." is never found. This is bad English use and makes section 4 very confusing to understand.

Violin Plots (Figures 5 and 8) are not a common and straight forward way to show the results and therefore is suggested to the authors to guide the reader how to interpret the plots. In addition, the mean and confidence intervals are not visible for the modified exponential function.

Reference to Appendix A for equation 1 is missing.

Theta_s is not included in table 1.

Picture of the physical experiment will enhance the paper readability significantly.

Double check the use of commas.

---

## Author Comment (AC1) · 26 Sep 2017

**1 Summary**

First, we want to thank both referees for their thorough reviews and diligent analysis of our work. They both pointed us towards several issues and helped us to greatly improve our work.

Both referees positively evaluate the main idea presented in the paper to develop a comparatively simple dual permeability model with few parameters.

However, the referees also point out three main points of criticism:

1. The model may be based on "questionable" soil physics
2. The need for the presented inversion scheme is not demonstrated and the chosen algorithms may not be the best choice for parameter optimization in the particular case of the presented model
3. The experiment may not be suited to the model as it may not represent the situation simulated by the model (no macropore flow, limited range of water content)

Below we sum up our response to these three points followed by a detailed answer to these and the remaining points in part 2 of this document.

**1.1 Revision of model formulas and soil physics**

We revised the model equations and corrected several formal and typing errors in some equations (eq. 2, eq. 4, eq. 14, eq. 15). We also inspected carefully our use of hydrological terms and changed several formulations that apparently were misleading or unclear.

Regarding the water retention functions, referee #1 proposed a change to the two exponential versions. We argue that these modifications are a worse approximation of the water retention function in the relevant moisture range. Despite, we will test the suggested changes and in addition will include additional retention functions with a steep decline to zero at matrix saturation in the analysis (see detailed comments for a discussion).

Regarding the infiltration formulation, we will compare results using one of the infiltration formulations suggested by referee #1, to the simplified current version. We will better clarify how the sharp front approach leads to the current formulation, for which referee #1 surmised that it confused hydraulic conductivity with infiltration capacity (see details).

Referee #1 criticized the neglecting of ET when calculating the vertical profile equations in section 4.1. ET is not ignored in the presented approach, but we will more clearly describe its handling.

Regarding the Boltzmann transformation, referee #1 criticized its usage for estimation of the hydraulic diffusivity function. We agree that several of the assumptions for this transformation were violated, but we are confident that the presented approach still delivers parameters of the hydraulic diffusivity function that are in the correct order of magnitude. The Boltzman-derived estimates are only required to provide starting values for the final optimization that does not make use of the Boltzmann transformation. However, given that the calibration procedure must be revised anyway and the use of the Boltzmann transformation is so controversial, we will try to set up the calibration scheme where a starting value of the diffusivity function parameter is obtained in another way.

**1.2 Inversion scheme**

In the revised version we will explicitly demonstrate convergence to local minima when starting with parameters not near the global optimum, a problem that caused us to spent much time to go to these more complicated steps in the inversion. Since we will change some of the model equations we will have

to revisit the inversion scheme and we will try to simplify it where possible. In particular we will try to remove the reliance on the Boltzmann transformation as it appears to be controversial in this case.

We will also try to use the suggested GLUE method for the optimization. However, given the fact the prior range of several parameters spans many orders of magnitude and the model runtime can be up to 10' seconds, the GLUE approach may be difficult to handle.

**1.3 Experiment**

We agree that the experiment was far from ideal design for the model validation. The reason is that we joined in to the experiment that was originally designed for experiments looking at herbivory effects on DOM fluxes.

However, we infer that there was macropore flow, given the almost immediate outflow observed after rain events on moderately dry soil for the two events that we used for model calibration. We demonstrated successful model calibration with suboptimal experimental data of a limited moisture range, and argue that with better experimental data the calibration would be better constrained.

However, since the use of this experiment was heavily criticized by referee #1 we decided to instead set up an "artificial experiment" with the model software MACRO 5.2 (Larsbo and Jarvis 2003). By using MACRO to create soil water content time series and outflow data we can make sure that macro pore flow exists in the data used to calibrate our model. We already did a calibration run with data received from MACRO and will include its results in the revised manuscript instead of results obtained using the mesocosm experiment.

**2 Detailed Discussion**

In the following we go through the Referee's replies and answer them paragraph by paragraph.

**A1 Referee #1 wrote:**

*"An experiment was carried out in which a soil with low clay content was sieved, homogenized and mixed with fine gravel. If one wished to avoid the development of a macropore network, this was the way to go. Grass was sown, which was well watered. Late in the paper it is stated that the roots were expected to create macropores, but they occupied these themselves. Dye tracer tests to verify after the experiment if there was preferential flow were not performed. The soil was kept moist so the chances of root shrinkage (opening up an air gap between the root and its macropore wall) were small. The shallow depth of the soil (20 cm) and the lower boundary with only a single outlet that presumably only permitted water to flow out at atmospheric pressure also ensured wet conditions throughout. Thus, the experiment allowed only a limited range of water content to be examined under the absence of macropores."*

As stated above we agree that the experiment was not optimal to test the model. We will not use data from the experiment in the revised manuscript. Instead we calculated "artificial" time series of water content using the model MACRO 5.2 (Larsbo and Jarvis 2003) and will present a calibration based on these data. MACRO is a widely used dual-permeability model with an explicit simulation of macropore flow. Therefore, we can make sure that macropore flow occurred in the data we are using for model calibration.

**A2 Referee #1 wrote:**

*"The expression for the flow velocity in the macropores is a factor 3 off. I presented an alternative derivation near Eq. (2). Eq. (4) is incorrect as well: the flux term in the mass balance is formulated incorrectly. I presented an improved version. The resulting version of the flow equation (the second part of Eq. (4)) is the same though. It appears that the error in the flow velocity and the mass balance term canceled out, perhaps that is why they went unnoticed."*

The missing factor of 3 was a typing error as can be seen from the correct formulation stated as equation (A9) in appendix A, which is identical to equation (5) of Hincapié and Germann (2009). The text

describing the derivation of equation (3) in paragraph 2.1 was incorrect. We provide a corrected version in the updated manuscript. We adopted the improved formulation of equation (4).

**A3 Referee #1 wrote:**
*"The conductivity at the top soil is falsely interpreted as the maximum infiltration rate (section 2.4). It is better to state that the maximum infiltration rate is determined by the hydraulic gradient and the hydraulic conductivity at the tip of the wetting front.*
*[…]*
*All in all, approaching matric infiltration based only on the conductivity at the soil surface, and assuming that is somehow a maximum flow rate is not very good. There are several infiltration equations (by Green-Ampt, Philip, and others) that should work better, especially in the early stages (e.g., Hillel, 1998, pp. 391-391)."*

Referee #1 is right in his explanation. The formulation used in our original manuscript is called the "sharp front approach" (Brutsaert 2005, section 9.4.2) and assumes that the top of the soil will become saturated very quickly during infiltration so that the potential gradient ($\Delta\psi$) becomes zero resulting in a maximum infiltration rate ($I_{max}$) that is equal to the hydraulic conductivity at the soil surface ($K_{top}$):

$$I_{max} = (1 + \Delta\psi)\, K_{top} = K_{top} \quad (1)$$

This approximation leads to an underestimation of infiltration at the very beginning of the rain event. We hypothesized that this error does not impair the simulation results, because of the shortness of this period compared to the entire simulation time (a few seconds vs. days).

In the revised manuscript we will state more clearly why we used this formulation. In addition we will compare our approach to the approach used in the current version of the MACRO model (5.2), which explicitly models the potential gradient based on equation (2) as (eq. 59 in Larsbo and Jarvis (2003)):

$$I_{max} = \left(1 + \left|\frac{\psi_{top} - \psi_1}{0.5\Delta z}\right|\right) K_{top} \quad (2)$$

where $\psi_{top}$ is the matric potential at the surface, which is 0 assuming an infinitesimal saturated boundary layer, $\psi_1$ is the matric potential at the center of the first grid cell and $\Delta z$ is the height of the first grid cell.

We want to avoid using explicit infiltration formulas such as the one by Green and Ampt, because these explicitly depend on time, which hampers more general application of the resulting numerical model.

In our original manuscript we calculated $K_{top}$ as the arithmetic mean of the saturated hydraulic conductivity and the hydraulic conductivity at the center of the first grid cell. This is identical to the way it is calculated in the MACRO model (Larsbo and Jarvis 2003). However, referee #1 pointed out that it may be better to use the geometric mean instead. Therefore, we did an optimization for each of the formulations based on the mesocosm data. As could be expected, the infiltration capacity predicted by the arithmetic mean was much higher than the prediction based on the geometric mean. However, both predictions behave in a similar manner and react with an exponential decline after the start of the precipitation event as is expected from infiltration theory. When the precipitation stops water still flows out of the topmost grid cell increasing infiltration capacity again.

So, clearly the choice of arithmetic vs. geometric mean does have an impact on the simulated infiltration and may therefore also influence the distribution of infiltrating water between macropores and soil matrix. However, while we think this is certainly worth further exploration – in particular since it also influences other dual-permeability models - we do not want to look into that in this paper and will therefore use the arithmetic mean as in MACRO.

**A4 Referee #1 wrote:**

*"In section 4.1, unit gradient conditions are confused with hydrostatic equilibrium. The authors claim that d theta / dz = 0 indicates hydrostatic equilibrium. But this is the expression for unit gradient flow, not that for hydrostatic equilibrium.*

*[…]*

*The explanation of Eq. (14) is simply false (and its sign is wrong). The expression for the matric potential follows directly from the definition of hydrostatic equilibrium (zero hydraulic gradient). Applying Eq. (8) for steady-state conditions is less specific (constant but non-zero flows are also steady state), and taking the derivative of the equation leads to additional terms that were conveniently ignored. In addition, the sign of the equation is only correct if z is defined positive upward, which contradicts Table 1. In Equation 15, the sign is wrong (a consequence from the error in Eq. 14). Apparently it escaped the attention of the authors that they calculate positive matric potentials in unsaturated soils with this equation."*

Again, referee #1 is right in his explanation. Following the referee's suggestion we use a more general derivation of equation 14:

$$\frac{d\psi}{dz} = 1 \ (3)$$

$$\frac{d\theta}{dz}\frac{d\psi}{d\theta} = 1 \ (4)$$

$$\frac{d\theta}{dz} = \left[\frac{d\psi}{d\theta}\right]^{-1} (5)$$

We will change this accordingly in the revised manuscript. As also pointed out by referee #1 equation 14 has the wrong sign in the original manuscript. This typo also carries over to equation (15). However, the vertical equilibrium profile functions given in Table 4 are still correct as they were derived using the correct versions of both equations.

**A5 Referee #1 wrote:**

*The casual statement in section 4 that ET had no effect even though a 20 cm closed container apparently was not at hydrostatic equilibrium (contrary to what the authors believe) also betrays a lack of understanding. Root water uptake tends to make soil moisture profiles in non-layered soils more uniform because the roots take up more water where it is easier to get.*

The approach presented in the original manuscript does not ignore ET. We calculate a constant rate of ET from the water content time series for the two depths where these are available and then do an interpolation over the whole soil profile (assuming a linear change between the two measurement depths and assigning the measured values to depths above and below the measurement depths, respectively). This results in a value for ET to be removed from every grid cell at every time step during the simulation. We will point this out clearer in the revised manuscript.

However, we do assume the influence of ET on the shape of the vertical equilibrium profile can be neglected and that therefore the profile depends on the matrix water retention properties once the outflow from the mesocosms stopped. This means that ET must not influence the vertical water content profile. This may happen for different reasons: E.g. water losses from the profile via ET could be of a similar fraction everywhere in the profile, or water losses through ET are very small and/or counterbalanced very fast so that the vertical water content profile is maintained in equilibrium and follows the matrix retention properties. We will make this clearer in the revised manuscript.

[Figure]

**Figure 1 – Matrix water retention functions. The simple exponential function (blue line) will slightly underestimate the true value of θs, but will predict values and have a slope similar to the van Genuchten function (black line) in an intermediate moisture range. The exponential function forced through $θ_s$ at ψ=0 (red line) as suggested by referee #1 has a higher slope at intermediate values of θ. We propose the revised exponential function (green line) which is close to the van Genuchten function.**

3  **A6 Referee #1 wrote:**

4  The exponential retention curves in Table 4 behave non-physically. The exponential terms in them go to 1 as theta goes to zero

5  *and to some small but finite value as theta goes to theta-s. If the exponent is made proportional to (theta-s minus theta) instead*

6  *of theta they behave as they should.*

8  Referee #1 suggests that the exponent of the two exponential retention functions should be changed to

9  be proportional to $(θ_s − θ)$. However, we do not think that this is necessary as the functions behave in a

10  meaningful manner:

11  The calculated potentials are always negative (using equation R1 and R2 in table 4 of the original

12  manuscript) as the exponent is multiplied by $ψ_0$ and $ψ_r$, respectively, both of which are (negative)

13  potentials.

14  While the exponential terms indeed approximate to one as θ goes to zero, the overall term becomes $ψ_0$

15  in case of the simple exponential equation (R1) and is not defined for the modified exponential equation

16  (R2) (but its absolute values become infinity for small values of θ). As θ increases both functions will give

17  smaller absolute numbers (the potential will be less negative) which is coherent with other, commonly

18  used retention functions and can also be seen in figure 4 of the manuscript, where the modified

19  exponential function behaves very similarly to the Campbell function. We agree, that the simple

20  exponential function will overestimate flow under very dry conditions, but on the other hand simulated

21  flow rates are very small at low moisture contents so that biases in equations at these values are not as

22  important.

23  However, one criticism remains: The calculated potential cannot become zero for both exponential

functions. Note, that this is also true for the Campbell and Brooks-Corey functions. However, water movement will not be too sensitive to these changes, given that the predicted potentials at matrix saturation are small for all functions.

Regardless, we want to explain our reasoning for including the simple exponential equation in our model. Assuming that the black lined van Genuchten retention function in figure 2 resembles reality the closest, a simple exponential equation (blue line) with a predicted potential different from 0 at $\theta_s$ can be tangential to the van Genuchten equation (and provide accurate estimates of potential) at intermediate water contents. However, this function will predict a value for $\theta_s$, which is smaller than the real value. In our model water exceeding this predicted value of $\theta_s$ will be in macropores, where it is not subject to matric potential. Accordingly, the model makes an error for water contents slightly above $\theta_s$.

The referee's suggestion to use an exponential function with an exponent proportional to $(\theta_s - \theta)$ would result in a function similar to the red line in figure 2. Forcing $\psi = 0$ at $\theta = \theta_s$ results in $\theta_s$ being closer to its real value and therefore minimizes above mentioned error. On the other hand the resulting function is no longer tangential to the van Genuchten function and will therefore give worse predictions of matrix potential for intermediate values of $\theta$.

To overcome this, we propose an additional two -parameter formulation:

$$\psi(\theta) = \psi_0\left(1 - e^{a(\theta_s - \theta)}\right) (6)$$

This formulation will result in $\psi = 0$ for $\theta = \theta_s$ and $\theta_s$ being close to its real value while still closely approximating the van Genuchten equation at intermediate values for $\theta$. We will add this function to our analysis in the revised manuscript.

**A7 Referee #1 wrote:**
*In section 4, the Boltzmann transformation is applied (Eq. 19). This approach is only useful when capillarity dominates gravity (thereby ensuring the validity of Eq. 18, where the gravity term is omitted) and conditions far away from the infiltration front have no effect because the transformed equation is solved for a semi-infinite column with a prescribed water content at infinity. The authors waited for 12 hours after rainfall and had only 20 cm long columns. The long wait diminished the matric potential gradients, thus ensuring that the gravitational gradient was no longer negligible. Also, the infiltration front had penetrated most if not all of the column by then, invalidating the semi-infinite approximation. The resulting values of the diffusivity are therefore meaningless. The fitted diffusivity parameters had an effect on the other parameters when in the last leg of the optimization process, all parameters were fitted simultaneously, but it is impossible to determine how detrimental they were. In any case, they cast doubt upon all fitted parameter values and the model runs based on these values.*

As already pointed out above, we agree that several of the assumptions for the Boltzmann transformation were violated in our approach. While we are confident that the presented approach still delivers a parameter for the hydraulic diffusivity function which is in the correct order of magnitude, we will try to change our calibration scheme to find a starting value for the diffusivity in another way.

The effect of the Boltzman transformation on the final results is probably small since it was merely used to find a starting value for one of the six parameters in the final optimization step.

**B1 Referee #2 wrote:**
*[…], the study points out that their aim is divided in two main parts; Developing the low complexity model and develop a model inversion scheme. For the former one, it is not clear to the reader or reviewer if lower complexity model is desirable as even the authors point out that the model can only be implemented for the specific case of the controlled conditions in the lab, the obtained results are not very good, they are only tested for one infiltration event of the 15 intended and last but not least, it reduces the flexibility for fitting the capacity of retention functions. For the latter part, it seems that the necessity of the inversion modeling process for the calibration is not sufficiently explained and seems merely a product of the availability of a software package. This last comment doesn't mean it cannot be done but it shouldn't be one of the main goals of the study as all models in any field require calibration and validation. If such a robust implementation is really required, the study should also  probe it*

*by comparing the results of different calibration methods so that the expected drawbacks (equability and local minima) are*
*proven to exist.*

The low complexity model is more desirable than a more complex model if it performs similar or even reasonably worse under the given constraints (only soil water content time series available for calibration). We will point out more clearly that the simpler model has the advantage to still work, when an inversion of a more complex model is impossible.

The current model implementation will require the specific conditions of the mesocosm experiment (or conditions close to this). However, some of the conditions can be easily changed (e.g. it is easy to adopt the model to different boundary conditions) and we plan to investigate further changes to the model to increase its applicability.

The referee is right in pointing out that it should not be the aim of our study to create a complicated parameterization scheme. We will improve our formulation in the revised manuscript. The aim of the study is rather to show that a calibration of our model is possible given the constraint of only having soil water content time series available. We ended up with the inversion scheme presented in the original manuscript since simpler schemes did not work. We will show this by example in the revised manuscript.

Since we will use output from the MACRO model instead of measurements from the mesocosms in the revised manuscript a further explanation of the available data and their usage will not be included. Therefore we want to explain the discrepancy between events measured and events used in the inversion here:

We conducted 15 sprinkling experiments in three mesocosms in parallel totaling 45 water content time series. Unfortunately there was an issue with the probes in one of the mesocosms, so we could only use time series from that mesocosm for its first two events (resulting in 32 time series). All of the remaining time series were used in the first inversion step estimating the vertical water content profiles and hydraulic diffusivity functions. Contrary the estimation of the macro pore parameters required events that exhibited outflow. This only left us with three time series from the second event as outflow only happened during the first two events and was not measured for the first event. These time series were used in all comparison plots.

**B2 Referee #2 wrote:**
*Regarding the model development section: Please explicitly include the reduction of parameters (which parameters are required if no simplification is done (presented Table 2) vs which parameter are required in the proposed model) with respect to the different methods so that the study becomes more attractive to the reader. It is important to clarify that the matric water diffusivity referred in the study is a "borrowed" term to refer to describe the gradient in time of the soil water content (in the differential form of a diffusion process) instead of a diffusive particle solute movement in gases or liquids. Hence it is recommended to use the term Hydraulic diffusivity as explained for example in the book of Hillel "Introduction to environmental Soil Physics".*

The revised manuscript will include a better explanation of parameter reduction and make clear where the simplifications happened. For this purpose we will use MACRO as a reference model. We also changed the manuscript to use the term "hydraulic diffusivity".

**B3 Referee #2 wrote:**
*The deduction of the macropore flux is missing a scheme (i.e. as presented by Hincapié and Germann 2009) in which its deduction and variables are presented for the new proposed model representation. The exchange lumped term r is a function the same 3 parameters (G,d and ) which are present in both Dual permeability and dual porosity models described in literature (See Gerke and Van Genuchten 1993, Evaluation of a first order water transfer term for variably saturated dual-porosity flow models). It would be interesting to compare the obtained results with the ones presented in literature as the posterior r distribution seems to reflect a high degree of uncertainty in the model. Furthermore, it would be valuable if this macropore exchange parameters where fixed based on literature and see how much they either reduce or increase the uncertainty bounds which will allow to have a first impression of their importance for the studied soil in the proposed model.*

We provided a derivation scheme for the macropore flux in appendix A. The revised manuscript will also include a comparison of the parameter r (which controls the exchange flux between both domains) with literature values. We will conduct a sensitivity analysis of this parameter, if its uncertainty remains high even when using the artificial time series data.

**B4 Referee #2 wrote:**
*According to the manuscript, 45 time series extracted from 15 rainfall events were measured but only 1 of them presented observable seepage fluxes which might imply that the experiment was not correctly designed. It might be related to the high silt content of the soil probes.*

See B1 for an explanation of the available data and their usage. We will not use data from the experiment in the revised manuscript.

**B5 Referee #2 wrote:**
*Also note that the method presented in Campbel (1974) was validated against sandy loam (higher content of sand than silt).*

We included this retention function since it only requires three parameters. As mentioned in the manuscript this function is a special case of the Brooks-Corey function with a residual water content of zero. For the texture of the mesocosms, the residual water content parameter can be close to zero and we therefore think the Campbell function will behave similar to the Brooks-Corey function. It is also just one of several functions used for comparison.

**B6 Referee #2 wrote:**
*Regarding the inverse modeling scheme: It is not explicitly mentioned which are the input parameters which are going to be found by the inverse modeling algorithm for each of the optimization steps. A very "robust" algorithm for inverse modeling is used for fitting a rather simple model (as pointed by the authors) with a reduced number of input parameters which gives the impression that its implementation might not be required. Two assumptions are pointed to support the use of the algorithm (e.g existence of local minimums and equifinality. For the former one, it must be proved that this is the case by implementing a simpler algorithm in which local minima is obtained. It is expected that with monotonous and continuous functions, the chances of finding local minima for the objective functions presented are reduced. For the latter one, it is expected that this is still the case as too many parameters (e.g. for the first inversion a„b, n and L,r,c and theta_s) are fitted from basically two measurements of soil water content (5 cm and 12 cm). All water retention functions are not linear which implies that at least 3 different measurements should be used for a reliable fitting*

In the revised manuscript we will clearly point out which parameters are to be found in each step of the inversion algorithm. As pointed out above we will also include evidence of convergence to suboptimal local minima with a simpler global optimization scheme. Even with the reduced complexity model there is still some equifinality given the water content data as shown by parameter correlations. The equifinality argument, though, does not contribute to choice of inversion scheme.
We use more than two data points in the inversions. For the global inversion we make use of the full water content time series with about 40 data points per time series.
For the same soil profile we fit data of several events and equilibrium profiles together and additionally make use of the experiment design in which events with outflow have a water content of $\theta_s$ at the lower model boundary shortly after outflow cedes. Essentially this gives us a third point per profile for these events.
As an example for 10 profiles we need to estimates two ($\theta_s$ is equal to water content after outflow at lower boundary) shape parameters that are the same among vertical profiles plus one reference water

content that differs across profiles (but is the same for all profiles with outflow), resulting in less than 2+10 estimated parameters versus at least 2*10 data points.

We will adopt this procedure to be used with the artificial data generated with MACRO. Since all vertical profiles share two of the parameters and some share the third, the amount of data points per profile used in the optimization is larger than 2.

**B7 Referee #2 wrote:**
*The uncertainty weights (w1 and w2) used inside the objective functions for minimization are chosen arbitrarily. Please explain the choices or even if possible, make a small sensitivity test as they can bias the obtained distributions for the unknown parameters.*

We set up the mesocosm experiment with three parallel treatments. Unfortunately the water content probes in one of the mesocosms failed shortly after the start of the experiment. Therefore we were left with no estimate of variation on the water content data. The measurement uncertainty given by the producer did not capture the variance across replicated events and mesocosms. Therefore we arbitrarily set the value to 10 % of the measured value. We will include an uncertainty analysis of the weights in the revised manuscript (which will – as mentioned – use the artificial data from MACRO).

**B8 Referee #2 wrote:**
*The authors imply that a double optimization inverse modeling framework is needed to avoid equifinality. So the questions that arise are: why are the authors want to avoid equifinality while including more than one water retention function as part of the study (Note that equifinality also implies that different models may perform similarly) and is it important to avoid it when the parameter results are presented as uncertainties for each parameter?*

We included several water retention functions in our analysis, since most of them are not widely used and we wanted to estimate, which degree of simplification is still reasonable. We do not point this out clearly in the original manuscript and will add it to the revised one.

Simplification was not done to avoid equifinality, but rather to avoid convergence to local minima. Equifiniality is a property of the model and the available data rather than of the inversion scheme. We do not want to imply that the inversion scheme avoids equifinality. Furthermore, reducing equifinaility is not required for the modeling purpose, although it helps to compare parameter posterior distributions to literature values.

**B9 Referee #2 wrote:**
*It will be very valuable to compare parameters obtained from the initial inverse modeling with respect to the same ones obtained in the next inverse modeling. Furthermore, if one of the main goals of the manuscript is to show the importance of the whole inversion method (Besides the simplification of the model as pointed in page 3, line 20), a comparison against a single inverse modeling optimization of all parameters will benefit the study significantly. Otherwise, what is the point of doing 2 step optimization without probing the effects of equifinality or local minima encountering ? As a suggestion: For this kind of studies, the GLUE (Beven and Binley) methodology may be a better option as it accepts the fact that equifinality is present for all paramters and models and hence evaluates the models based on a likelihood function.*

We will add a comparison of parameter distributions between the two inversions to the manuscript. As pointed out above we developed the complicated inversion scheme, since a simple inversion did not work and will include results of a simpler inversion attempt in the revised manuscript.

We will look into the GLUE methodology, but it may prove to be difficult to handle given the fact that the prior range of several parameters spans many orders of magnitude and the model runtime can be up to 10' seconds.

**B10 Referee #2 wrote:**

*Results and discussion: In figure 3, the range of measurements differs from the ones obtained for the estimated functions. If these curves were used as seed values for the next optimization, would it imply that the measurements obtained at -5 cm are useless? Otherwise explain the implication of not having the fitted functions inside the measured bounds. Moreover, is it necessary to perform the initial model inversion? Ideally this step will avoid the entrapment on local minima but evidence of such value must be supported, especially for the fitting on monotonic functions such as the ones presented in this manuscript.*

The initial estimate of the vertical water content profiles hits both data ranges well. (not shown) . Figure 3, contrary, uses the parameters fitted to the time series in the last inversion step (without any equilibrium assumption) and predicts the equilibrium profile. Fig. 3 suggests that water is only close to equilibrium at the times for which the data is displayed. We will show a comparison of parameters after the first and last inversion step that shows the slight discrepancy introduced by the equilibrium assumption in the first inversion step. The first step only delivers approximate starting values of a subset of the parameters.

In the revised manuscript, we will also show examples of failing inversions if the initial inversion step is omitted.

**B11 Referee #2 wrote:**

*In figures 3 and 4, it is not clear how the 90% confidence intervals where obtained and they are almost non-visible, especially in figure 4. In Figure 4, it is also not clear why the dashed line (Shiozawa) and long dashed line (Oosterveld and Chang) cut the horizontal axis between 0.2 and 0.3 theta values. Whereas the predicted values may go as far as 1.0.*

The uncertainty estimates were calculated from a Markov Chain Monte Carlo (MCMC) run starting at the optimal parameter values. We will add an explanation to the figure labels.

The predictions of the pedotransfer functions may not be very accurate (that is why we used three of them in the first place), since they depend on soil data bases with a limited scope that may not be representative for the mesocosm soils. Also, the exponential retention functions used in this study overestimate matrix potential at low moisture contents (see discussion of retention functions above).

The revised manuscript will not include the pedotransfer functions, since we will compare the predicted retention functions with the van-Genuchten function as parameterized by the MACRO model.

Note that the exponential retention functions are only plotted up to $\theta_s$ in Figure 4 of the original manuscript at which point they jump to $\psi = 0$.

**B12 Referee #2 wrote:**

*In section 6.1 it is stated that there was a significant reduction of uncertainty after the second model inversion step but it is also acknowledged that data from the first optimization is not shown. Why? What would be the result with one general inversion modeling of all the parameters at the same time?*

We will add a comparison of parameter distributions and results between the two inversions to the manuscript.

**B13 Referee #2 wrote:**

*In the last paragraph of the conclusions it is stated that the prediction of the water content time series at 12 cm is somewhat worse than the prediction at 5 cm. However, the parameter estimation is better fitted for the measurements at 12 cm than the ones obtained at 5 cm which also are said to be close to the global optimum (see Figure 3). But appreciations would mean that the author is either getting good results at 5 cm with worst fitting values at this depth and is getting bad results in the moisture content time series at 12 cm while having a good fit of the retention function at that depth. This may be a good evidence that one inverse modeling procedure of all parameters at the same time might be better (but more complicated) than making a first optimization for the water retention functions as a not so "optimal" and restrictive*

Indeed, the model reproduces the water content time series at 5 cm depth better than at 12 cm depth. However, the reviewer-stated better fit to 12cm is a misunderstanding of Fig 3 as described above (B10). Note that all displayed results referred to the last inversion step, which is the proposed inversion of all the parameters.

**B14 Referee #2 wrote:**

*Soil heterogeneity and uncertainty derived from the initial conditions and its implications in the model is not further discussed in the manuscript.*

We will add this to the discussion in the revised manuscript as applicable.

**B15 Referee #2 wrote:**
*Minor comments: […]*

We found the minor comments very helpful and will follow their advice.

**Other changes**

We will add below sketch to the manuscript to better illustrate the concept of kinematic wave flow in macropores.

[Figure]

1    References

3    Brutsaert, W. (2005): Hydrology. Cambridge University Press.

4    Hincapié, I.A. and Germann, P.F. (2009): Abstraction from Infiltrating Water Content Waves during Weak

5    Viscous Flows. Vadose Zone Journal, 8, 4, 996 – 1003.

6    Larsbo, M. and Jarvis, N. (2003): MACRO 5.0. A model for water flow and solute transport in

7    macroporous soil. Technical description. Emergo Studies in the Biogeophysical Environment. Swedish

8    University of Agrucultural Sciences.